

# Recent advances in the inverse design of silicon photonic devices and related platforms using deep generative models

Sun Jae Baek and Minhyeok Lee

Department of Intelligent Semiconductor Engineering, Chung-Ang University, Seoul, Republic of South Korea

## ABSTRACT

This article presents an overview of recent research on the inverse design of optical devices using deep generative models. The increasing complexity of modern optical devices necessitates advanced design methodologies that can efficiently navigate vast parameter spaces and generate novel, high-performance structures. Established optimization methods, such as adjoint and topology optimization, have successfully addressed many design challenges. However, the increasing complexity of modern optical devices creates opportunities for complementary approaches. Deep generative models offer additional capabilities by leveraging their ability to learn complex patterns and generate novel designs. This review examines various deep learning methodologies, including multi-layer perceptrons (MLP), convolutional neural networks (CNN), auto-encoders (AE), Generative Adversarial Networks (GAN), and reinforcement learning (RL) approaches. We analyze their applications in the inverse design of photonic devices, comparing their effectiveness and integration in the design process. Our findings indicate that while MLP-based methods were commonly used in early research, recent studies have increasingly employed CNN, GAN, AE, and RL methods, as well as advanced MLP models. Each of these methods offers unique advantages and presents specific challenges in the context of optical device inverse design. This review critically evaluates these deep learning-based inverse design technologies, highlighting their strengths and limitations in the context of optical device design. By synthesizing current research and identifying key trends, this article aims to guide future developments in the application of deep generative models for optical device inverse design.

## INTRODUCTION

Optical devices are sophisticated instruments that utilize the properties of light for various functions, including control and information processing (*Saleh & Teich, 2019*; *Okamoto, 2021*; *Goutzoulis, 2021*; *Moore & Smart, 2020*; *Borrelli, 2017*). These devices can be classified according to their light-matter interactions, which determine their operational principles and design approaches (*Molesky et al., 2018*). Some devices, such as LEDs and lasers, generate optical output at specific wavelengths (*Schubert, 2018*; *Liu et al., 2020*;

Corresponding author
Minhyeok Lee, mlee@cau.ac.kr

*Qiao et al., 2019*). Others manipulate light by altering its path, reflecting it, or controlling its polarization, such as 3D lenses and TVs (*Gao et al., 2017*; *Ou, Li & Tang, 2016*). Additionally, sensors detect light for various purposes, from measuring intensity to analyzing spectral components (*Ming Qing et al., 2020*; *Kim et al., 2021*; *Javaid et al., 2021*).

Optical devices play a crucial role in numerous sectors, including defense (*Chandrasekar et al., 2019*), medicine (*Bobokulova, 2023*), culture (*Kim et al., 2019*), telecommunications (*Anzalchi, Inigo & Roy, 2017*), semiconductor manufacturing (*Dey, 2018*), display technology (*Chen et al., 2024*), and sensor development (*Zhu et al., 2019*). Their importance continues to grow as their applications expand and contribute to technological advancements (*Mentzer, 2017*).

Traditionally, the design of optical devices involves the creation of initial prototypes and then experimentally evaluating their functionality (*Wang et al., 2019*; *Salehi et al., 2019*). This approach allows for the development of mathematical models based on fundamental physical principles, including variables such as the angle of incidence, angle of reflection, and refractive index of light. These models enable accurate predictions of optical device properties (*Sujecki, 2018*; *Jiang, Chen & Fan, 2021*). However, this method is often time-consuming and complex, making it challenging to develop devices with novel structural designs (*Bogaerts & Chrostowski, 2018*; *Casellas et al., 2024*).

To address these limitations, researchers have turned to inverse design methodologies (*Kang et al., 2024a*; *Khaireh-Walieh et al., 2023*). In this approach, the desired outcomes guide the design process, using computational algorithms to systematically explore and define optimal configurations (*Wiecha et al., 2021*; *Mao et al., 2021*). This method reduces the reliance on physical prototypes and increases the potential to discover innovative solutions that conventional approaches might overlook (*Butt, Khonina & Kazanskiy, 2021*; *Ren et al., 2021*; *Capmany & Pérez, 2020*).

The field of optical device design has seen significant advances through various optimization techniques. Notably, adjoint optimization and topology optimization methods have demonstrated remarkable success in addressing complex design challenges (*Chung & Miller, 2020a*, *2020b*). These approaches have become fundamental tools in the inverse design of optical devices, offering efficient solutions for many applications. Recent work has also shown promising results in combining these traditional optimization methods with neural networks (*Kang et al., 2024b*), suggesting potential synergies between different approaches.

The integration of artificial intelligence (AI) into the inverse design of nanophotonic devices offers a promising strategy to further improve the design process (*Jackson, 2019*; *Zhang & Lu, 2021*; *Berente et al., 2021*). AI is expected to streamline the workflow, accelerate the development of optical devices with novel structures and potentially initiate significant changes in the field (*Khailany, 2020*). This study explores the application of deep learning models to rapidly design optical devices with desired characteristics, in order to enhance the efficiency of the design process beyond that of conventional methodologies (*Deng et al., 2022*; *Verganti, Vendraminelli & Iansiti, 2020*).

Previous reviews have examined various aspects of deep learning in nanophotonic design. For instance, *Liu et al. (2021)* classified deep learning architectures into three main

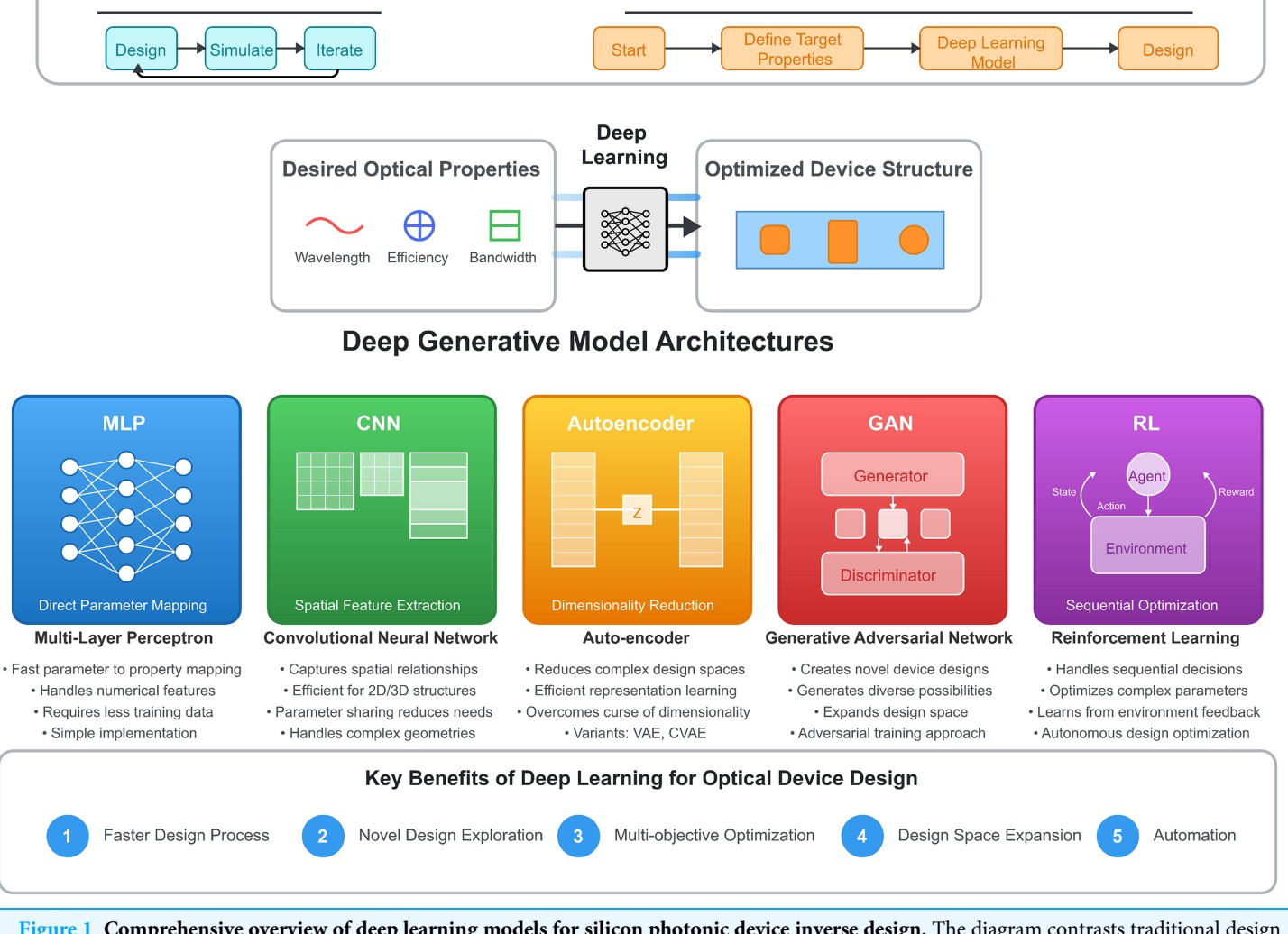

**Figure 1** **Comprehensive overview of deep learning models for silicon photonic device inverse design.** The diagram contrasts traditional design (iterative forward approach) with deep learning-based inverse design (starting from desired properties). The central section illustrates the inverse design workflow, while the lower section details the distinct architectural characteristics of each model type: multi-layer perceptron (MLP) for direct parameter mapping, convolutional neural network (CNN) for spatial feature extraction, autoencoder (AE) for dimensionality reduction, Generative Adversarial Network (GAN) for novel design generation, and reinforcement learning (RL) for sequential optimization. Each approach offers unique advantages in addressing specific challenges in silicon photonic device design.

categories (multi-layer perceptron (MLP), convolutional neural network (CNN), and recurrent neural network (RNN)) and focused primarily on inverse design for metasurfaces. However, their work did not include more recent advancements in AI, such as Transformer (*Vaswani et al., 2017*) and diffusion models (*Ho, Jain & Abbeel, 2020*). Similarly, the research conducted by *Pan & Pan (2023)* on silicon optical devices lacked

specificity regarding the deep learning methods employed in particular contexts, as detailed categories for deep artificial neural networks were not defined.

Given the rapid advancement of deep learning methodologies and the expanding scope of inverse design in photonics, this review aims to provide an overview of recent developments in deep learning methods applied to the inverse design of nanophotonic devices. We systematically categorize the methods according to the specific situations where each deep learning technique was applied, offering insights into the current state of the field and potential future directions. Building on the fundamental context and motivations, Fig. 1 provides a systematic visual overview of the deep learning models explored in this review. The diagram illustrates the paradigm shift from traditional forward design to inverse design approaches, along with the distinct architectural characteristics of each deep generative model (MLP, CNN, autoencoder (AE), Generative Adversarial Network (GAN), and reinforcement learning (RL)). Each model offers unique capabilities for addressing specific challenges in silicon photonic device design, as will be explored in the following sections. This visual framework helps clarify how these diverse approaches contribute to the advancement of inverse design methodologies.

## LITERATURE ANALYSIS

### Purpose of reviews

The inverse design of optical devices using deep generative models has been applied across diverse platforms including metamaterials, optical fibers, and quantum devices. This review specifically focuses on silicon-based nanophotonic devices. This deliberate scope allows us to provide an in-depth analysis of the unique challenges and solutions in silicon photonics, which has become increasingly important for integrated optical circuits and telecommunications. Silicon photonics offers distinct advantages such as complementary metal-oxide-semiconductor (CMOS) compatibility, high integration density, and established fabrication processes, making it a crucial platform for next-generation optical devices. By maintaining this focused scope, we aim to offer detailed insights that complement existing reviews covering other optical platforms, contributing to the advancement of silicon-based nanophotonic device design methodologies. This review aims to provide a comprehensive examination of recent trends in inverse design for nanophotonic devices using deep learning methods. We analyze relevant studies that employ various deep learning models to enhance our understanding of inverse design processes for nanophotonic devices. Additionally, we identify the strengths and limitations of previous research in this field, providing guidance for future studies on inverse design for nanophotonic devices using deep learning.

### Survey methodology

The survey methodology for this review was conducted in three main stages, designed to ensure a comprehensive and relevant collection of studies. First, we selected the Web of Science (WOS) database as our primary source of peer-reviewed articles. WOS was chosen for its extensive collection of carefully edited and peer-reviewed scholarly journals, which provides a high level of academic rigor to our research. This database selection was crucial

in ensuring that our review was based on high-quality, peer-reviewed research in the field of nanophotonic device inverse design. Our article follows systematic review methodology, which requires transparent documentation of literature search and selection processes to ensure reproducibility. This approach allows other researchers to validate our methodology and build upon our systematic analysis of deep learning applications in optical device design.

For the second stage, we performed a keyword search using the terms "deep learning", "inverse design", and "nanophotonic devices". This combination of keywords was specifically chosen to target studies at the intersection of artificial intelligence and nanophotonic design. The initial search yielded 216 articles, which we then refined through manual screening. We included articles focusing on nanophotonic devices composed of chemical components, as these represent the core of our research interest. Studies on circuit-related devices were excluded to maintain focus on optical systems. Additionally, we excluded articles that solely discussed optimization techniques during the inverse design process, as our primary interest was in the application of deep learning methods to the entire inverse design workflow.

Lastly, we limited our review to studies published between 2019 and 2024. This 5-year time frame was selected to capture the most recent advances in inverse design for nanophotonic devices, reflecting the rapid progress in deep learning technology during this period. By focusing on this recent timespan, we aimed to provide an up-to-date analysis of the field, capturing the latest methodologies and innovations. This approach ensures that our review offers insights into the current state of the art and potential future directions in the application of deep learning to nanophotonic device inverse design.

Table 1 provides an expanded overview of the deep learning methods used in optical device design, as found in the reviewed articles. The deep learning methods presented in Table 1 represent the primary approaches used in the inverse design of optical devices. Each method offers unique advantages in addressing specific challenges in optical device design. MLPs provide a straightforward approach for mapping between device parameters and optical properties, making them suitable for rapid performance prediction. CNNs excel in processing spatial data, making them particularly useful for analyzing and generating device geometries. AEs offer powerful dimensionality reduction capabilities, enabling efficient representation of complex device structures. GANs have shown promise in generating novel device designs that meet specific performance criteria, potentially leading to innovative solutions. RL approaches the design process as a sequential decision-making problem, which can be particularly effective for optimizing complex, multi-parameter device structures.

The reviewed articles demonstrate a broad distribution across various journals, highlighting the widespread international interest and engagement in this research field. This interdisciplinary spread provides strong evidence of the appeal of inverse design for nanophotonic devices using deep learning methodologies. As illustrated in Fig. 2A, more than half of the reviewed articles were published within the last 3 years, indicating the rapid growth and current relevance of this research area. Figure 2B shows the citation distribution of the selected articles. While some articles received fewer than ten citations,

**Table 1 Expanded overview of deep learning methods for optical devices design.**

| Methods | Brief description | Application in silicon based optical device design | References |
|---|---|---|---|
| MLP | A feedforward neural network structure with multiple hidden layers. Each layer consists of nodes that process and transmit information to the next layer. This learns by adjusting weights to minimize errors *via* backpropagation. | Used for mapping between device parameters and optical properties, enabling rapid prediction of device performance. Optimizes device geometries by correlating structural features with desired optical characteristics. | *Taud & Mas (2018)*, *Liu et al. (2022, 2023)*, *Kojima et al. (2021)*, *Head & Keshavarz Hedayati (2022)*, *Ma & Li (2020)*, *Chen et al. (2023)*, *Jiang & Fan (2020)*, *Guo et al. (2022)*, *Ren et al. (2021)*, *Gao et al. (2019)* |
| CNN | Alternates between convolution and pooling layers to extract and reduce image features. Particularly effective for processing grid-like data. | Applied to analyze and generate 2D or 3D structures of optical devices, capturing spatial relationships in device geometries. | *Alzubaidi et al. (2021)*, *Gostimirovic et al. (2023)*, *Song et al. (2021, 2020)*, *Chen et al. (2022)*, *Shi et al. (2022)*, *Ma et al. (2022)* |
| AE | Compresses and reconstructs data, facilitating efficient feature learning and dimensionality reduction. Consists of an encoder and a decoder. | Employed for dimensionality reduction of complex optical device structures, enabling efficient representation and manipulation of device designs. | *Bank, Koenigstein & Giryes (2023)*, *Li et al. (2022)*, *Kiarashinejad, Abdollahramezani & Adibi (2020)*, *Hong & Nicholls (2022)*, *Tang et al. (2020)*, *Zhu et al. (2023)* |
| GAN | Generators and Discriminators compete to improve the ability to generate new data. The generator creates synthetic data while the discriminator evaluates its authenticity. | Used to generate novel optical device designs that meet specific performance criteria, expanding the design space beyond conventional methods. | *Aggarwal, Mittal & Battineni (2021)*, *Kim et al. (2022)*, *Dizaji, Habibiyan & Arabalibeik (2022)* |
| RL | Agent learns to execute tasks based on the environment and implements policies to maximize rewards. Involves iterative learning through interaction with an environment. | Applied in optimizing device structures through sequential decision-making processes, particularly useful for complex, multi-parameter optimization problems in optical device design. | *Sutton & Barto (2018)*, *Jiang & Yoshie (2022)*, *Zhao et al. (2022, 2023)*, *Sajedian, Badloe & Rho (2019)*, *Hwang, Lee & Seok (2022)* |

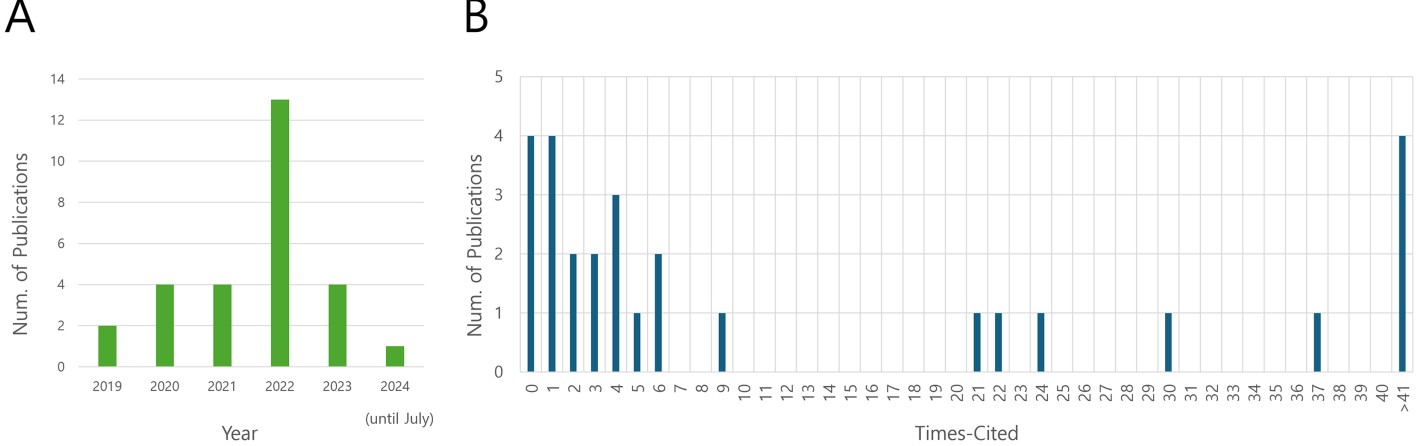

**Figure 2 Overview of the distribution of publication years and citation frequencies.** (A) Distribution of publication years; (B) Distribution of citation frequencies.

over 30% of the selected articles garnered more than twenty citations, suggesting their significant influence in the field. This citation pattern underscores the impact and importance of recent research in the inverse design of nanophotonic devices using deep learning approaches.

**Table 2 Detailed distribution of publications on recent optical devices inverse design in deep learning.**

| Publication | Counts | Percentage (%) | Key focus areas |
| --- | --- | --- | --- |
| Optics Express | 3 | 9.3 | Nanophotonic structures, metamaterials, optical waveguides |
| Nature Photonics | 2 | 6.2 | Nano structures, photonic structures |
| Laser & Photonics Reviews | 2 | 6.2 | Photonic integrated circuits, metamaterials |
| Nanophotonics | 2 | 6.2 | Plasmonic devices, metasurfaces |
| Journal of Lightwave Technology | 2 | 6.2 | Integrated photonics, fiber optics |
| ACS Photonics | 2 | 6.2 | Nanoscale optics, quantum photonics |
| IEEE Photonics Journal | 2 | 6.2 | Photonic devices, optical communications |
| Misc | 17 | 53.1 | Various topics including photonic crystals, optical sensors, and nonlinear optics |
| Total | 32 | 100 | |

## Distribution of publications

Table 2 presents a detailed distribution of publications on recent optical devices inverse design using deep learning. The distribution of publications presented in Table 2 offers valuable insights into the current research trends of inverse design for optical devices using deep learning. Optics Express has emerged as the leading journal in this field, accounting for 9.3% of the reviewed articles. The prominence of this journal likely stems from its broad scope in optics and its rapid publication process, which is particularly advantageous in the fast-evolving field of deep learning applications.

Six other journals, Nature Photonics, Laser & Photonics Reviews, Nanophotonics, Journal of Lightwave Technology, ACS Photonics, and IEEE Photonics Journal, each contribute 6.2% of the reviewed articles. This even distribution across these specialized journals indicates the multifaceted nature of the research, spanning various sub-domains of photonics and optical device design. Nature Photonics covers breakthrough research in optical science and technologies, with particular emphasis on novel photonic structures and nano structures. Laser & Photonics Reviews focuses on comprehensive reviews in laser physics and photonics applications, especially in emerging fields like metamaterials and Photonic integrated circuits. Nanophotonics focuses on the interaction of light with nanoscale structures, while the Journal of Lightwave Technology emphasizes applications in optical communication systems. ACS Photonics covers a broad range of topics in photonics with a particular focus on nanoscale phenomena, and IEEE Photonics Journal provides a platform for rapid dissemination of original research in photonics.

Notably, the majority of articles (53.1%) are distributed across various other journals. This wide distribution underscores the interdisciplinary nature of the field, with research findings published in journals spanning physics, materials science, electrical engineering, and computer science. It also suggests that the application of deep learning to the inverse design of optical devices is of interest to a broad scientific community, not limited to traditional journals in optics and photonics.

The diversity in the venues for publication reflects the multifaceted nature of the research field, which combines expertise from optical physics, device engineering, and

artificial intelligence. This interdisciplinary approach is crucial for advancing the field as it allows for the integration of domain knowledge in optics with cutting-edge machine learning techniques. Furthermore, the distribution of publications indicates that while there are a few journals that have published multiple articles in this field, there is no single dominant venue. This suggests that the research community in this area is still evolving, with contributions from various scientific communities. It also highlights the potential for further specialized venues or special issues focused on the intersection of deep learning and optical device design to emerge in the future. With this comprehensive understanding of the publication landscape, we can now examine in detail how specific deep generative models have been applied to optical device inverse design, beginning with MLPs as the foundational architecture.

# DEEP GENERATIVE MODELS FOR OPTICAL DEVICE INVERSE DESIGN

## Multi-layer perceptron

MLP (*Taud & Mas, 2018*) is a fundamental deep learning architecture that has found significant applications in the inverse design of optical devices. This section explores the structure of MLPs and their various implementations in nanophotonic design.

### Structure and functionality of MLPs

An MLP consists of multiple layers of interconnected neurons organized in a feedforward manner. The network typically comprises an input layer, one or more hidden layers, and an output layer. Each neuron in a layer is connected to every neuron in the subsequent layer, allowing for complex data transformations. The input layer receives the initial data, hidden layers perform nonlinear transformations, and the output layer produces the final predictions or classifications.

Figure 3 illustrates the typical configuration of an MLP used in the inverse design of optical devices. This network structure facilitates the discovery of complex relationships between input parameters and output optical characteristics, making it particularly suitable for mapping between device parameters and optical properties.

### Applications in nanophotonic design

MLPs have been extensively applied in the inverse design of silicon-based nanophotonics, particularly for identifying appropriate nanostructure parameters (*Liu et al., 2022*, *2023*; *Kojima et al., 2021*; *Head & Keshavarz Hedayati, 2022*; *Ma & Li, 2020*; *Chen et al., 2023*; *Jiang & Fan, 2020*; *Guo et al., 2022*; *Ren et al., 2021*; *Gao et al., 2019*).

*Quantum nanoparticle shell design*

*Liu et al. (2022)* employed a fully connected neural network approach for the inverse design of quantum nanoparticle shells with multilayer configurations. The MLP architecture implemented in this article consists of an input layer with two nodes representing the period and filling ratio, followed by three hidden layers containing 512, 256, and 128 nodes respectively, and the output layer comprises 101 nodes, which predict the electromagnetic energy distribution and light-matter interaction strength at different

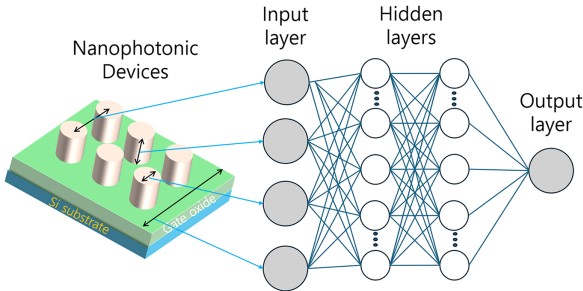

**Figure 3 Schematic representation of a MLP for optical device inverse design.**

frequencies. This hierarchical structure was carefully designed to effectively capture the complex relationships between geometric parameters and optical responses. Their work demonstrated the potential for precise manipulation of frequency and amplitude in quantum optics. However, they observed an increasing error rate as the thickness of the designed shells increased, highlighting a limitation of their approach.

Building on this work, *Liu et al. (2023)* addressed the shell thickness issue by designing a system that also influences the electromagnetic environment. Their approach used a fully connected neural network-based inverse design process, advancing the field by considering the interplay between quantum systems and the electromagnetic environment produced by nanophotonic structures.

*Material and spectral property correlation*
Extending the methodology beyond nanoscale parameter design, *So, Mun & Rho (2019)* developed a neural network that correlates information on the materials forming each layer of the shell with the electric and magnetic dipole spectra. The implemented MLP architecture features an input layer with four nodes corresponding to core radius, shell thickness, core permittivity, and shell permittivity. The network comprises four hidden layers, each containing 100 nodes, followed by a single output node predicting the resonant wavelength. This symmetric architecture was selected to maintain consistent feature extraction capability throughout the network depth. This approach enables the inverse design of shell nanophotonics to achieve desired spectral properties within specific shell layers in a matter of seconds, representing a significant advancement in design speed and precision.

*Silicon-based nanophotonic applications*
In practical applications of silicon-based nanophotonics, MLPs have been used for the structural design of various devices. For instance, *Kojima et al. (2021)* tackled the inverse design of a power splitter for enhancing microwave outputs. They implemented a MLP architecture comprising four hidden layers with 50 nodes each for silicon grating coupler design. The network processes three geometric input parameters such as depth, period and width. The output layer predicts transmission and reflection coefficients at wavelength. They overcame the challenge of requiring substantial training data by using a neural

network comprised of fully connected layers, randomly generating designs to optimize performance towards desired objectives.

*Head & Keshavarz Hedayati (2022)* applied an MLP network to the inverse design of distributed Bragg reflectors, which manipulate light paths using stacked thin film materials. They developed a architecture with three hidden layers for distributed Bragg reflector design. The network processes twenty input parameters representing thicknesses of ten alternating dielectric layer pairs. Outputs spectral reflection data across a broad wavelength range in the visible spectrum, with a high number of spectral points. Their goal was to achieve targeted reflection spectra through inverse design of layer thicknesses. To address the challenge of insufficient training data limiting the generation of specific colors, they implemented a tandem neural network combining an MLP architecture with a forward network. This approach allowed them to produce targeted spectral outcomes through inverse design.

*Tandem neural networks and optimization techniques*
Tandem neural networks combines two MLP. For example, the first MLP extracts laser parameters from experimental data, while the second MLP performs inverse design by predicting optimal device parameters for desired laser performance characteristics. The limitation of tandem neural networks in addressing only single-solution problems has been addressed by integrating particle swarm optimization techniques. *Ma & Li (2020)* combined a tandem neural network with particle swarm optimization, representing a heuristic-based optimization approach. Their architecture with three hidden layers for semiconductor laser parameter extraction. The network processes four input parameters such as active layer thickness, carrier density, optical intensity and injection current. This method expanded the search process from local to global, effectively resolving the single-solution limitation of tandem neural networks in the inverse design of semiconductor lasers.

*Chen et al. (2023)* further advanced the application of tandem neural networks by introducing a high-speed solution that addressed the issue of extended inverse design simulation times. They designed a network with four hidden layers of 256 nodes each for photonic crystal waveguide optimization. The network accepts six geometric parameters representing the size of two circular holes and how far these holes are shifted from their original positions, to predict how different frequencies of light will travel through the structure. Their tandem network-based regression neural network resolved data class imbalance issues and established a foundation for optimal parameter identification.

*Adaptive architecture and multi-objective optimization*
*Jiang & Fan (2020)* demonstrated an innovative approach using ResNet-based architectures that incorporates four MLP layers with 512 nodes each within residual blocks for photonic structure optimization. Their method featured an adaptive depth variation from deep to shallower architectures as the optimization progressed, enabling a more focused search for solutions needed in inverse design. This approach was successfully applied to the design of thin-film stacks composed of multiple material types, achieving

global optimization significantly faster than traditional algorithms used in photonic design.

*Efficient learning and parameter optimization*

Addressing the computational costs in the learning process, *Guo et al. (2022)* introduced the concept of an extendable neural network. An extendable neural network is an architecture that can add new layers while preserving previously learned knowledge. Unlike conventional MLP that requires complete retraining, this structure allows network expansion to learn new design conditions while maintaining existing performance. They introduced an extendable neural network architecture consisting of a base network with three hidden layers and two extension layers for nanophotonic waveguide design. Their innovation was implementing an extendable architecture that could adapt to new design constraints without retraining the entire network. This approach significantly reduces computational time when design requirements change and the database needs to be updated, followed by retraining, contributing to more efficient parameter optimization in the inverse design process.

*Genetic algorithm integration*

To address the issue of requiring large amounts of labeled data in nanophotonic device design, *Ren et al. (2021)* proposed a genetic algorithm-based MLP model. They suggested a genetic algorithm-optimized MLP with three hidden layers for silicon photonic polarizer design. Their optimization goal was to maximize polarization selectivity while maintaining high transmission. Unlike conventional MLP models that rely solely on backpropagation during training, this method optimizes the geometric structure of photonic devices in a polar coordinate system, even with relatively limited training datasets.

*Color production in nanophotonics*

In the context of achieving high resolution in silicon-based nanophotonics, *Gao et al. (2019)* employed a tandem neural network architecture for the intricate design of nanophotonics capable of producing specific colors. They developed a bidirectional MLP architecture featuring two parallel networks with four hidden layers for silicon color design. One network accepts geometric parameter to predict reflection spectra, while the inverse network transforms desired RGB colors into structural parameters. This approach optimized the inverse design process for producing over a million colors in nanophotonics, reducing costs by avoiding situations where different geometric structures might yield the same color output. However, they noted that increasing model accuracy requires a larger database of nanophotonics representing specific colors, highlighting a trade-off between accuracy and computational resources.

### Limitations and future directions

The application of MLPs in the inverse design of optical devices has shown significant progress, but several limitations and areas for future research have emerged from the reviewed studies. A primary challenge is the scalability of MLP models as the complexity of

optical devices increases. As observed in the work of *Liu et al. (2022)*, the error rate increased with growing shell thickness in quantum nanoparticle designs. This indicates that as device structures become more intricate, the computational demands for MLP-based inverse design grow substantially. Future research could focus on developing more efficient network architectures or training methods to address this scalability issue, potentially incorporating techniques from other deep learning domains such as sparse or quantized neural networks.

The generalization capability of MLP models in optical device design remains a significant concern. Current models often perform well within the range of their training data but may struggle with extrapolation to new design spaces. This limitation was evident in the study by *Head & Keshavarz Hedayati (2022)*, where insufficient training data restricted the generation of specific colors in distributed Bragg reflectors. Improving the generalization capabilities of these models is crucial for their practical application in diverse optical design scenarios. An effective approach would be integrating physics-informed neural networks, which incorporate known physical laws into the learning process to enhance the model's ability to generalize beyond the training data.

The integration of MLPs with multi-physics simulations represents another important direction for future research. Many optical devices operate in complex environments involving multiple physical phenomena. The work of *Liu et al. (2023)* on designing systems that influence the electromagnetic environment of quantum systems highlights the importance of considering multiple physical aspects in the design process. Future research could focus on developing hybrid models that combine MLPs with first-principles simulations to create more robust and realistic inverse designs.

While MLPs have significantly reduced design times compared to traditional methods, there is still room for improvement in achieving real-time inverse design capabilities, particularly for complex, multi-parameter systems. The tandem neural network approach proposed by *Chen et al. (2023)* represents a step towards high-speed solutions, but further advancements are needed. Future work could explore the use of reinforcement learning techniques or meta-learning approaches to enable rapid adaptation of MLP models to new design requirements, potentially leading to near-real-time inverse design capabilities.

Analyzing the trends in the reviewed articles, we observe a shift from single-objective to multi-objective optimization in optical device design. The work of *Jiang & Fan (2020)* on adaptive ResNet-based architectures for multi-objective optimization exemplifies this trend. Future research is likely to continue this trajectory, developing more sophisticated multi-objective optimization techniques for optical device design. Another emerging trend is the integration of MLPs with other optimization techniques, as seen in the work of *Ma & Li (2020)*, who combined tandem neural networks with particle swarm optimization. This hybrid approach addresses some limitations of pure MLP-based methods and opens up new possibilities for global optimization in complex design spaces. Future research could explore other combinations of MLPs with evolutionary algorithms or Bayesian optimization techniques to further enhance the inverse design process. The trend towards more efficient learning and parameter optimization, as demonstrated by *Guo et al. (2022)* with their extendable neural network concept, is likely to continue. Future work in this

direction could focus on developing more adaptive learning strategies that can efficiently update MLP models as design requirements change, potentially incorporating concepts from continual learning or transfer learning in deep neural networks. Additionally, the application of MLPs in nanophotonic color production, as explored by *Gao et al. (2019)*, highlights the potential for MLPs in highly specific and precision-demanding applications. Future research could extend this approach to other areas of nanophotonics where precise control over optical properties is crucial, such as in the design of metasurfaces or photonic crystals for specific light manipulation tasks.

MLPs provide a strong foundation for inverse design. However, the spatial nature of optical devices naturally directs attention to CNN that are specifically designed to handle structural and spatial relationships.

## Convolutional neural network

CNNs have emerged as a powerful tool in the field of artificial intelligence, particularly for tasks involving visual recognition and classification (*Alzubaidi et al., 2021*). Their application in the inverse design of optical devices has shown significant promise, especially in the domains of nanophotonics and photonic crystal design. This section explores the structure, capabilities, and applications of CNNs in the context of optical device inverse design.

### Structure and functionality of CNNs

The architecture of a CNN typically consists of several distinct layers: convolutional layers, pooling layers, and fully connected layers. This structure is designed to efficiently extract and process spatial features from input data, making CNNs particularly well-suited for analyzing visual information. Figure 4 illustrates the general structure of a CNN used in the inverse design of optical devices.

In the context of optical device design, CNNs offer several advantages over traditional MLPs. While MLPs often struggle with handling spatial information, CNNs excel in tasks where spatial relationships are crucial. This capability makes CNNs particularly effective for complex structural design in nanophotonics, where the spatial arrangement of components significantly influences device performance.

### Applications in nanophotonic design

The application of CNN architectures in the inverse design of silicon-based nanophotonics has been demonstrated at both the process level for feedback and the design level for simulation scenarios. Several recent studies have showcased the effectiveness of CNNs in various aspects of nanophotonic design (*Gostimirovic et al., 2023*; *Song et al., 2021*, *2020*; *Chen et al., 2022*; *Shi et al., 2022*; *Ma et al., 2022*). CNNs have enabled several key advances in silicon-based nanophotonic design. At the process level, *Gostimirovic et al. (2023)* developed a CNN-based method that automatically corrects fabrication-induced structural variations using small-scale microscope images, improving dimensional accuracy. For design optimization, *Song et al. (2021)* utilized CNNs to efficiently handle wavelength and polarization parameters in power splitter design, reducing computational overhead

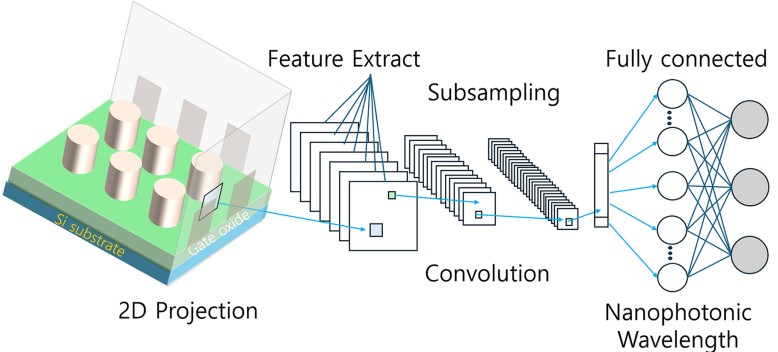

**Figure 4 Schematic representation of a CNN for optical device inverse design.**

compared to traditional MLPs. *Chen et al. (2022)* achieved breakthrough performance in electromagnetic field prediction using a U-Net architecture (WaveY-Net), enabling rapid evaluation of device performance without full simulations. Additionally, *Shi et al. (2022)* demonstrated the potential of capsule networks to achieve comparable results with only 60% of typical training data requirements, addressing the critical challenge of data efficiency in nanophotonic design.

*Process-level design correction*
*Gostimirovic et al. (2023)* addressed a significant challenge in nanophotonic manufacturing: structural deviations arising from fabrication processes. They proposed a process-level CNN-based inverse design method that automatically corrects design layouts before fabrication. This innovative approach employs pre-trained CNN predictive models to adjust the design layout using only small-scale microscope images. By implementing various corrective actions, this method aligns the fabricated structures more closely with the original designs, potentially reducing manufacturing errors and improving device performance.

*Efficient inverse design of power splitters*
In the design of silicon-based nanophotonic devices, *Song et al. (2021)* identified a limitation in conventional MLP approaches when dealing with wavelength and polarization parameters. They found that introducing these parameters into data-driven models significantly increased computational costs. To address this issue, they leveraged the dimensionality reduction capabilities of CNN models. This approach enhanced computational efficiency in the inverse design process, allowing for more complex parameter spaces to be explored without prohibitive computational overhead.

*Wavelength-dependent device design*
Recognizing the need for controllers that can manage diverse parameter information comprehensively, *Song et al. (2020)* proposed a method for inverse designing nanophotonic devices that incorporates wavelength controllers. This approach allows for the comprehensive consideration of both temporal and spectral characteristics. Their

CNN-based method predicts controllable wavelengths, implementing both forward prediction and inverse design. By predicting values that meet targeted goals across different wavelengths and energy densities, this method offers a more holistic approach to device design, considering the wavelength-dependent behavior of optical devices.

*High-accuracy electromagnetic field prediction*
*Chen et al. (2022)* utilized CNN architectures to predict electromagnetic field distributions with high accuracy and speed. They employed the U-Net architecture, which they termed WaveY-Net, in the inverse design process. This approach allowed for the determination of geometric structures of devices, including device height and material refractive indices. The ability to rapidly and accurately predict electromagnetic field distributions is crucial for the efficient design of complex nanophotonic devices, as it allows designers to quickly evaluate the performance of potential designs without resorting to time-consuming full-field simulations.

*Capsule neural networks for spatial hierarchies*
Moving beyond traditional CNNs, *Shi et al. (2022)* explored the use of capsule neural networks in nanophotonic design. Capsule networks are designed to effectively handle spatial hierarchical structures, making them suitable for tasks that require understanding complex spatial information in nanophotonic structures. The concept of capsules allows for the expression of various properties and the capture of relationships within hierarchical spatial information. This approach addresses some limitations of traditional CNNs, such as difficulties with pose variations and spatial transformations. Notably, capsule networks demonstrated the potential to achieve performance comparable to CNN architectures using only 60% of the training data, suggesting their capability to efficiently inverse design more complex nanophotonic structures with a greater number of design variables.

*DenseNet for enhanced learning capacity*
Further showcasing the capacity of CNN-based architectures to assist in high-accuracy inverse design processes with limited data, *Ma et al. (2022)* introduced the concept of DenseNet to nanophotonic design. DenseNets incorporate densely connected networks that more effectively learn complex patterns and features from data, enhancing the network's learning capacity. This approach is particularly valuable when dealing with limited training data, a common challenge in nanophotonic design where generating large datasets can be computationally expensive or experimentally challenging.

### Limitations and future directions
The CNNs have shown promise in optical device inverse design, but several challenges remain. A key issue is the quality and quantity of training data. As noted by *Ma et al. (2022)*, biases or errors in the original training data can significantly impact nanoantenna design. This underscores the need for high-quality, diverse datasets in nanophotonic design. Future studies could investigate efficient data generation techniques or methods to augment existing datasets while maintaining physical accuracy.

The interpretability of CNN models in optical device design is another concern. While CNNs demonstrate high predictive accuracy, interpreting how they arrive at their predictions remains a challenge. Scientists then use these predictions to make informed design decisions about optical devices. The challenge of interpreting complex, high-dimensional models, which is a limitation common to all data-driven methods discussed in this review, can affect their adoption in scientific applications where understanding underlying physics is crucial. Research into attention mechanisms or layer-wise relevance propagation techniques could provide insights into CNN decision-making in optical design tasks.

Generalizing new design spaces not represented in training data is a critical challenge. *Gostimirovic et al. (2023)* showed CNNs' potential to adapt to manufacturing variations, but extending this adaptability to novel design concepts remains an open question. Investigating transfer learning or meta-learning approaches could address this limitation, allowing CNN models to more effectively generalize to new optical device designs.

The computational resources required for training and deploying large CNN models, especially for complex nanophotonic structures, present another obstacle. While *Song et al. (2021)* demonstrated CNNs' efficiency in handling high-dimensional parameter spaces, further optimization of CNN architectures for optical design tasks is necessary. Research into more compact CNN architectures or hardware-specific optimizations could reduce computational overhead.

Current trends show a shift towards specialized CNN architectures for specific aspects of optical device design. *Chen et al.*'s *(2022)* work on U-Net architecture for electromagnetic field prediction and *Shi et al.*'s *(2022)* use of capsule networks for handling spatial hierarchies exemplify this trend. Future research may continue this trajectory, developing increasingly specialized CNN architectures for different optical design problems. Integration of CNNs with other computational techniques is another emerging trend. *Song et al.*'s *(2020)* combination of CNNs with wavelength controllers suggests a move towards more comprehensive design approaches. Future work could explore the integration of CNNs with multi-physics simulations or other machine learning techniques to create more holistic inverse design frameworks. The trend towards data-efficient CNN models, as seen in *Shi et al.*'s *(2022)* work with capsule networks achieving comparable performance using only 60% of training data, is likely to continue. This direction is particularly important in nanophotonic design, where generating large datasets can be computationally expensive. Future research could focus on developing more data-efficient CNN architectures or exploring few-shot learning techniques.

Several promising directions for future CNN-based inverse design research emerge. Integrating physics-informed constraints into CNN architectures could produce more physically accurate and reliable designs. This approach could combine CNN efficiency with physics-based model reliability, potentially addressing interpretability concerns. Advancements in multi-scale modeling using CNNs could enable more comprehensive optimization of optical devices. Developing CNN architectures that simultaneously handle multiple spatial scales would allow for optimization of both nanoscale features and overall device geometry, potentially leading to more efficient optical designs. Incorporating

uncertainty quantification techniques into CNN-based inverse design methods represents another essential research avenue. Providing uncertainty estimates in generated designs could offer valuable information about their reliability and robustness, which is crucial for practical applications in optical device manufacturing.

Furthermore, advancing CNN-based methods towards real-time inverse design capabilities could revolutionize optical device design. While current methods have reduced design times compared to traditional approaches, achieving proper real-time design optimization remains challenging. Research into more efficient CNN architectures, possibly combined with reinforcement learning techniques, could pave the way for rapid prototyping and on-the-fly optimization of optical devices. After exploring architectures designed for spatial pattern recognition, this section examines Auto-encoder, which provide unique capabilities for dimensionality reduction such as a critical aspect for handling the complexity of optical device design spaces.

## Auto-encoder

The AEs represent a significant advancement in the application of deep learning techniques to the inverse design of optical devices. These neural network architectures, as described by *Bank, Koenigstein & Giryes (2023)*, are designed to compress input data into a low-dimensional latent space and subsequently reconstruct it back into the original high-dimensional space. This capability makes AEs particularly suitable for handling the complex, high-dimensional data often encountered in nanophotonic optical device design.

### Structure and functionality of AEs

The structure of an AE typically consists of two main components: an encoder and a decoder. The encoder compresses the input data into a low-dimensional representation, while the decoder attempts to reconstruct the original input from this compressed form. Figure 5 illustrates the configuration of an AE used in the inverse design of optical devices.

In the context of nanophotonic design, AEs serve a crucial role in dimensionality reduction and feature extraction. They are particularly effective in identifying the most salient design variables and reducing the dimensionality of the design space, thereby enhancing the efficiency of the inverse design process. This capability makes AEs well-suited for finding input variables corresponding to target outputs, a key requirement in inverse design problems (*Li et al., 2022*; *Kiarashinejad, Abdollahramezani & Adibi, 2020*; *Hong & Nicholls, 2022*; *Tang et al., 2020*; *Zhu et al., 2023*).

### Mathematical formulation and specific impact of AEs

The encoder $f_\theta$ that maps the input data $x$ to a latent representation $z = f_\theta(x)$, and the decoder $g_\phi$ that reconstructs the input from this latent representation $\hat{x} = g_\phi(z)$. Accordingly, the mathematical framework of the AE can be formulated as follows:

$$L_{AE}(\theta, \phi) = \frac{1}{n} \sum_{i=1}^{n} \left|\left| x_i - g_\phi(f_\theta(x_i)) \right|\right|^2.$$

This mathematical framework offers several specific advantages.

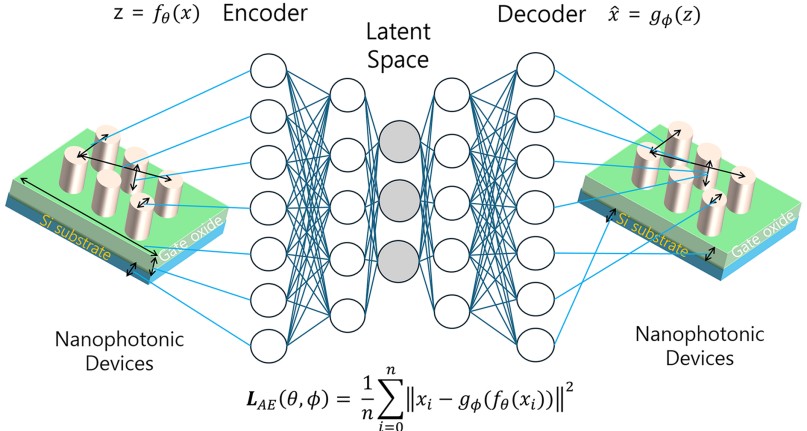

$$L_{AE}(\theta, \phi) = \frac{1}{n}\sum_{i=0}^{n}\left\| x_i - g_\phi(f_\theta(x_i)) \right\|^2$$

**Figure 5 Schematic representation of an AE for optical device inverse design.**

*Dimensionality reduction for design space exploration*
As demonstrated by *Li et al. (2022)*, the encoder maps high-dimensional device structures to a significantly lower-dimensional latent space. This compression facilitates efficient design space exploration, mathematically represented as follows:

$$z = f_\theta(x) \quad \text{where} \quad z \in \mathbb{R}^k, x \in \mathbb{R}^d, k \ll d.$$

*Manifold learning for valid designs*
The decoder $g_\phi$ learns to map points from the latent space back to physically realizable device structures, effectively learning the manifold of valid optical designs. *Kiarashinejad, Abdollahramezani & Adibi (2020)* utilized this property to transform multi-to-one inverse design problems into more tractable one-to-one mappings.

$$\hat{x} = g_\phi(z) \quad \text{where} \quad \hat{x} \approx x \quad \text{for all valid designs}$$

*Bayesian optimization in latent space*
*Li et al. (2022)* leveraged the compressed latent representation for efficient Bayesian optimization. Rather than optimizing in the original high-dimensional space, they performed optimization in the latent space.

$$z^* = \arg\max_z \mathscr{F}(g_\phi(z)) \text{ where } \mathscr{F} \text{ represents the desired optical performance metrics.}$$

The optimal device design is then obtained as $x^* = g_\phi(z^*)$.

For example, in the unidirectional transmission nanophotonics design by *Li et al. (2022)*, the AE compressed a design space with hundreds of geometric parameters into a 20-dimensional latent space. This 95% reduction in dimensionality allowed their Bayesian multi-sampling approach to efficiently identify designs achieving 95% light passage in one direction while blocking over 60% in the opposite direction. Similarly, *Tang et al. (2020)* applied a CVAE to the inverse design of nanophotonic power splitters. Their mathematical

formulation conditioned the latent space on desired transmission spectra, allowing for targeted design generation.

$$p_\phi(x|y) = \int p_\phi(x|z,y) p_\theta(z|y) dz.$$

where $y$ represents the target optical spectrum, $z$ is the latent variable, and $x$ is the device structure. This formulation enabled direct mapping from performance specifications to optimal device geometries, demonstrating how the dimensionality reduction capabilities of AEs translate to practical advances in optical device design.

### Addressing the curse of dimensionality

One of the primary challenges in nanophotonic design is the curse of dimensionality, which occurs when the increase in data dimensions leads to an exponential growth in the amount of data required for effective analysis. This phenomenon can significantly degrade the performance of neural networks. AEs offer a powerful solution to this problem by effectively reducing the dimensionality of the design space.

Li et al. (2022) demonstrated the effectiveness of AEs in circumventing the curse of dimensionality. They introduced an unsupervised learning-based inverse design method that utilizes Bayesian multi-sampling to handle high-dimensional data without evaluating all possible design scenarios. This approach significantly reduced the computational resources required for the inverse design process. The researchers applied this method to design unidirectional transmission nanophotonics, achieving 95% light passage in one direction while blocking over 60% in the opposite direction. However, they noted that verification of the actual performance of these manufactured nanostructures was lacking, highlighting an area for future research.

Building on theoretical simulations, Kiarashinejad, Abdollahramezani & Adibi (2020) extended the application of AEs to actual sample fabrication. They utilized AEs to reduce the dimensions of both design and response spaces in electromagnetic (EM) nanophotonics. This approach simplified multi-to-one design problems into more manageable one-to-one problems, demonstrating the practical applicability of AEs in nanophotonic design. However, the researchers identified potential technical issues when integrating this approach with conventional EM analysis software packages, indicating a need for further development in this area.

### Variations of auto-encoders

Recent research has explored variations of the traditional AE model to enhance its capabilities in nanophotonic design. Two notable variations are the variational auto-encoder (VAE) and the conditional variational auto-encoder (CVAE).

#### Variational auto-encoder

Hong & Nicholls (2022) employed the VAE model to address the curse of dimensionality. VAEs enhance the latent space learning capability by filtering out confusing or irrelevant

data. The researchers proposed a method that facilitates the generation of new data samples closely resembling the original data through simple encoding and decoding processes. This approach has shown significant computational efficiency advantages in the design process of thin-film optical films, demonstrating the potential of VAEs in nanophotonic design.

*Conditional variational auto-encoder*
*Tang et al. (2020)* introduced the application of CVAEs to complex physical photonic device design. Previously used in medical image synthesis and cybersecurity, this study marked the first application of CVAEs in the context of nanophotonic design. The researchers demonstrated the effectiveness of CVAEs in designing nanophotonic power splitters. However, they noted that the generalizability of the CVAE model to other types of nanophotonics remains uncertain, highlighting an area for future investigation.

Further advancing the application of CVAEs, *Zhu et al. (2023)* introduced a hybrid optimization algorithm combined with a generative model using CVAE. This method encodes and decodes the spectral responses and features of designed nanofilms, compressing the matching actual spectrum into the latent space. The researchers demonstrated superior performance of the CVAE-based inverse design approach compared to alternatives, further establishing the potential of this technique in nanophotonic design.

### Limitations and future directions
The AEs have shown potential in nanophotonic inverse design, but several challenges persist. A primary issue is the interpretability of latent space representations. While AEs excel at dimensionality reduction, as demonstrated by *Li et al. (2022)* in their work on unidirectional transmission nanophotonics, the physical meaning of compressed representations often remains unclear. This lack of clarity can impede the practical application of AE-based designs, especially in scientific contexts where understanding underlying physical principles is crucial. Future studies could explore methods to map latent space dimensions to specific physical parameters of optical devices, potentially through integrating physics-based constraints in the AE architecture.

The ability of AE models to generalize to novel design spaces not represented in the training data is another concern. This limitation was evident in the study by *Tang et al. (2020)*, where the generalizability of their CVAE model to other types of nanophotonics beyond power splitters was uncertain. Enhancing the extrapolation capabilities of AE models to new design regimes is essential for their broader application. Research into transfer learning techniques or developing more robust latent space representations that capture fundamental physical principles could address this issue.

Integrating AE-based approaches with conventional electromagnetic analysis software presents technical challenges, as noted by *Kiarashinejad, Abdollahramezani & Adibi (2020)* in their work on dimensionality reduction in electromagnetic nanophotonics. This integration is critical for practically adopting AE-based inverse design methods in engineering workflows. Future work could focus on developing standardized interfaces or

middleware solutions that facilitate seamless communication between AE models and established simulation tools.

A significant limitation highlighted by *Li et al. (2022)* is the lack of experimental validation for AE-designed nanostructures. While their theoretical approach achieved impressive performance in unidirectional light transmission, the actual fabrication and testing of these designs were not reported. This gap between theoretical design and practical implementation represents a significant challenge. Future studies should prioritize the experimental realization of AE-designed optical devices, including the development of fabrication techniques capable of realizing the complex structures generated by these models.

Current trends indicate a shift towards more sophisticated AE variants for specific aspects of optical device design. The progression from basic AEs to VAEs and CVAEs, as seen in the works of *Hong & Nicholls (2022)* and *Zhu et al. (2023)*, illustrates this trend. These advanced models offer enhanced capabilities in generating new design samples and handling conditional design requirements. Future research may continue this trajectory, developing increasingly specialized AE architectures that can capture unique characteristics and constraints of different optical design problems.

The integration of AEs with other optimization techniques is another emerging trend. The hybrid optimization algorithm combined with a CVAE-based generative model, proposed by *Zhu et al. (2023)*, exemplifies this approach. This direction suggests a move towards more comprehensive inverse design frameworks that leverage the strengths of multiple techniques. Future work could explore the combination of AEs with gradient-based optimization methods or evolutionary algorithms to create more powerful and flexible inverse design tools.

The trend towards addressing the curse of dimensionality in nanophotonic design, as demonstrated by *Li et al. (2022)* and *Kiarashinejad, Abdollahramezani & Adibi (2020)*, is likely to remain a key focus. As the complexity of optical devices increases, developing more efficient dimensionality reduction techniques will be crucial. Future research could investigate advanced manifold learning techniques or hierarchical AE structures that can effectively capture multi-scale features in optical device designs.

Several promising directions for future research in AE-based inverse design of optical devices emerge. The development of physics-informed AEs represents a particularly promising avenue. By incorporating known physical laws and constraints into the AE architecture, these models could produce more physically realistic and feasible designs. This approach could bridge the gap between data-driven and physics-based modeling, potentially leading to more robust and reliable inverse design methods.

Extending AE-based approaches to handle multi-objective optimization tasks is another important direction. Many practical optical design problems involve trade-offs between multiple performance criteria. Developing AE architectures capable of navigating these complex multi-dimensional optimization landscapes could enable the design of optical devices with unprecedented combinations of properties.

The application of AEs to inverse design problems involving dynamic or reconfigurable optical devices represents an unexplored frontier. Most current studies focus on static

device structures, but many advanced applications require devices that can adapt to changing conditions. Developing AE models that can capture and optimize the temporal behavior of optical devices could open new possibilities in fields such as adaptive optics or programmable photonics. AE models excel at efficient representation learning. However, addressing the need for novel design generation necessitates an examination of GANs, which have demonstrated exceptional capabilities in producing new and optimized optical device designs.

### Interpretability and practical realization

The black-box nature of AE latent spaces presents significant challenges for physical implementation. Unlike traditional design methods where each parameter has clear physical meaning (like the width of a waveguide directly affecting mode confinement), AE latent variables often lack intuitive physical interpretation. This is similar to how a complex recipe might be compressed into a few key instructions, while efficient, important details about individual ingredients may be lost.

Several approaches can improve the interpretability and practical realization of AE-based designs:

- **Disentangled representations:** Developing AE architectures where individual latent dimensions correspond to specific physical properties, similar to how musical notation separates pitch, duration, and volume into distinct parameters.
- **Physical parameter mapping:** Creating mapping functions between latent variables and physical parameters, acting as a "translation dictionary" between the compressed representation and fabrication specifications.
- **Fabrication simulation feedback:** Incorporating simulated fabrication effects into the training process, allowing the model to learn which design features are most sensitive to manufacturing variations.

Recent experimental implementations have begun to address these challenges. For instance, *Li et al. (2022)* demonstrated theoretically promising unidirectional light transmission devices, but their work also highlighted the need for experimental verification and systematic approaches to translate latent space representations into fabrication-ready designs.

## Generative adversarial network

GANs have emerged as a powerful tool in the inverse design of optical devices, offering unique capabilities in generating novel nanophotonic structures. As described by *Aggarwal, Mittal & Battineni (2021)*, the GAN model comprises two primary components: a generator and a discriminator, which operate in an adversarial process to learn from data.

### Structure and functionality of GANs

The GAN architecture is based on a game-theoretic approach, utilizing the minimax algorithm. The generator aims to produce synthetic data that closely resembles authentic data, while the discriminator's role is to distinguish between real and synthetic samples.

This adversarial interplay drives both components to improve continuously: the generator strives to create increasingly realistic data, while the discriminator hones its ability to detect subtle differences between real and generated samples.

Figure 6 illustrates the configuration of a GAN used in the inverse design of optical devices. This setup is particularly advantageous for creating novel nanophotonic designs that do not exist in current datasets, making it a valuable tool for pushing the boundaries of optical device design.

### Applications in nanophotonic design

The application of GANs in nanophotonic design has shown significant promise, particularly in generating 2D structures with specific optical properties. Several recent studies have demonstrated the effectiveness of GAN-based approaches in this field (*Kim et al., 2022*; *Dizaji, Habibiyan & Arabalibeik, 2022*).

*Maximizing transmittance at specific wavelengths*
*Kim et al. (2022)* proposed a GAN-based inverse design method aimed at achieving maximum transmittance at specific wavelengths. Their approach addressed a common challenge in GAN training: the stability of the adversarial process. To mitigate this issue, they designed the discriminator to be deliberately weaker than the generator, promoting a more stable training process. This strategy allowed for the generation of nanophotonic devices with desired characteristics more reliably. Furthermore, *Kim et al. (2022)* introduced the concept of conditional GANs with an added control vector and a controllable classifier. This innovation enabled the generation of nanophotonic devices with specific, desired characteristics. By incorporating these conditional elements, their method offered greater control over the generated designs, allowing for more targeted and application-specific outcomes.

*Inverse design of miniaturized spectrometers*
In a related study, *Dizaji, Habibiyan & Arabalibeik (2022)* applied GAN-based techniques to the inverse design of miniaturized spectrometers. Spectrometers play a crucial role in accurate spectral analysis across various fields, including pharmaceuticals, food science, and materials engineering. The miniaturization of these devices presents significant challenges, making it an ideal application for advanced inverse design methods. *Dizaji, Habibiyan & Arabalibeik (2022)* employed a two-step approach in their design process. First, they used an AE to identify the optimal spectrum of interaction between filters and light. This step helped in defining the target characteristics for the spectrometer design. Subsequently, they demonstrated the effectiveness of using a GAN-based neural network to physically design the optical components that would produce the optimal spectrum identified by the AE.

This hybrid approach, combining AEs and GANs, showcases the potential for integrating different deep learning architectures in the inverse design process. By leveraging the strengths of both AEs (in feature extraction and dimensionality reduction) and GANs (in generating novel designs), *Dizaji, Habibiyan & Arabalibeik (2022)* were able to address the complex challenge of spectrometer miniaturization effectively.

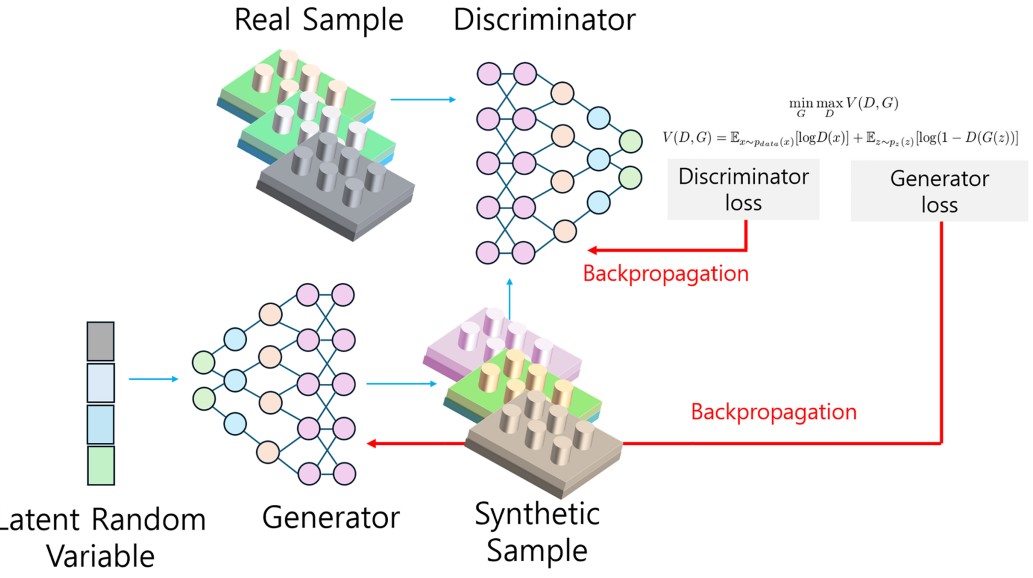

**Figure 6 Schematic representation of a GAN for optical device inverse design.**

### *Limitations and future directions*

The GANs have demonstrated potential in optical device inverse design, yet several challenges persist. A primary issue is ensuring the physical realizability of generated structures. While GANs excel at creating novel designs, as shown by *Kim et al. (2022)* in their work on maximizing transmittance at specific wavelengths, translating these designs into manufacturable structures remains problematic. This limitation highlights the need to directly incorporate physical constraints and fabrication considerations into the GAN architecture. Future research could focus on developing physics-informed GANs that integrate known physical laws and manufacturing limitations into the generative process.

While GANs offer powerful generative capabilities for nanophotonic design, they face several inherent challenges that affect their practical implementation. Training instability remains a significant obstacle, often manifesting as mode collapse where the generator produces limited design varieties, or oscillating behavior where the model fails to converge. These issues are particularly problematic in nanophotonic applications where design diversity is essential for exploring novel solutions. Advanced GAN variants such as Wasserstein GAN (*Arjovsky, Chintala & Bottou, 2017*) could address these stability concerns by using an alternative loss function that provides more reliable gradients during training. Additionally, GANs frequently generate artifacts or physically unrealizable structures that, while mathematically satisfying the optimization criteria, violate manufacturing constraints or electromagnetic principles. Incorporating physics-based constraints directly into the adversarial training process could significantly improve the practicality of GAN-generated designs. Future research should explore these hybrid approaches that combine the creative potential of GANs with domain-specific knowledge to ensure both innovative and manufacturable nanophotonic structures.

The stability of GAN training in optical device design contexts presents another significant challenge. *Kim et al. (2022)* addressed this by deliberately designing the discriminator to be weaker than the generator, but this approach may limit overall GAN performance. Developing robust training techniques specifically for optical design applications is crucial for broader GAN adoption in this field. Future work could explore adaptive training strategies that dynamically adjust the balance between the generator and discriminator based on target optical property complexity.

Current GAN applications in optical device design primarily focus on 2D nanophotonic structures, as seen in both reviewed studies. Extending GAN-based approaches to 3D optical structure design represents a crucial next step. This extension would allow for more complex and versatile optical device creation but also introduces significant challenges in computational complexity and data representation. Future research could investigate novel GAN architectures capable of efficiently generating and optimizing 3D nanophotonic structures.

The validation of GAN-generated designs through experimental testing remains a critical challenge. While both *Kim et al. (2022)* and *Dizaji, Habibiyan & Arabalibeik (2022)* demonstrated promising results in simulation, practical implementation and testing of these designs were not fully explored. Bridging this gap between computational design and experimental validation is essential for practically adopting GAN-based inverse design methods. Future studies should prioritize the fabrication and testing of GAN-designed optical devices, including developing feedback loops that can inform and improve GAN models based on experimental results.

Analyzing trends in the reviewed articles, we observe a shift towards more targeted and conditional GAN architectures. The work of *Kim et al. (2022)* on conditional GANs with added control vectors represents this trend, allowing for more precise control over generated nanophotonic structures. This direction suggests a move towards more application-specific GAN models in optical device design. Future research is likely to continue this trajectory, developing increasingly specialized GAN architectures that can capture and optimize specific optical properties or device functionalities.

Another emerging trend is the integration of GANs with other deep learning techniques, as exemplified by *Dizaji, Habibiyan & Arabalibeik (2022)* in their combination of autoencoders and GANs for spectrometer design. This hybrid approach leverages the strengths of multiple techniques to address complex design challenges. Future work could explore more sophisticated combinations of GANs with other machine learning models, such as reinforcement learning for optimizing device performance or graph neural networks for capturing complex structural relationships in optical devices.

Several promising directions for future research in GAN-based inverse design of optical devices emerge. The development of multi-objective GAN models represents a particularly important avenue. Many practical optical design problems involve trade-offs between multiple performance criteria. Creating GAN architectures capable of simultaneously optimizing multiple optical properties could lead to more versatile and efficient design processes. This could involve development of novel loss functions that balance different

optical characteristics or the creation of GAN ensembles that specialize in different aspects of device performance.

Incorporating advanced electromagnetic simulation techniques directly into the GAN training process is another crucial direction. While current approaches often rely on simplified physical models or *post-hoc* simulation, integrating more sophisticated electromagnetic solvers into the GAN architecture could lead to more accurate and reliable design outputs. This integration presents significant challenges regarding computational efficiency and differentiability, but it could dramatically improve the practicality of GAN-generated designs.

Applying GANs to dynamic or tunable optical devices represents an unexplored frontier in the field. Most current studies focus on static device structures, but many advanced applications require devices that can adapt to changing conditions. Developing GAN models that can generate designs for reconfigurable or actively tunable optical devices could open new possibilities in fields such as adaptive optics or programmable photonics. Lastly, addressing the interpretability of GAN-generated designs remains a significant challenge. While GANs can produce effective designs, explaining why these designs work often proves challenging. Enhancing the interpretability of GAN-generated optical devices is essential for building trust in these methods among the scientific community. Future research could investigate techniques for extracting physical insights from trained GAN models. Specifically, convolutional attention mechanisms could help identify which spatial features of the device structure most strongly influence particular optical properties. Additionally, incorporating graph attention networks could provide interpretable representations of how different components in the nanophotonic structure interact to produce desired optical behaviors.

### Manufacturing challenges and practical implementation

While GANs show theoretical promise, their practical implementation faces specific manufacturing challenges. The complex, often non-intuitive structures generated by GANs may include features that are difficult to fabricate using current nanofabrication techniques. Think of GAN-generated designs as intricate sculptures that might look perfect in a digital environment but contain details too fine for real-world tools to carve. For example, high-aspect-ratio features (very tall, thin structures) commonly appearing in GAN outputs often collapse during fabrication due to mechanical instability, similar to how a tall, thin clay structure might collapse before firing.

To address these challenges, several practical approaches can be implemented:

- **Fabrication-aware constraints:** Integrating manufacturing rules directly into the GAN training process, similar to how a sculptor must consider the properties of their material. For instance, minimum feature size limitations (typically 10–100 nm depending on the fabrication technique) can be encoded as penalties in the loss function.
- **Two-stage optimization:** First generating an ideal design, then refining it through a secondary clean-up algorithm that smooths problematic features while preserving

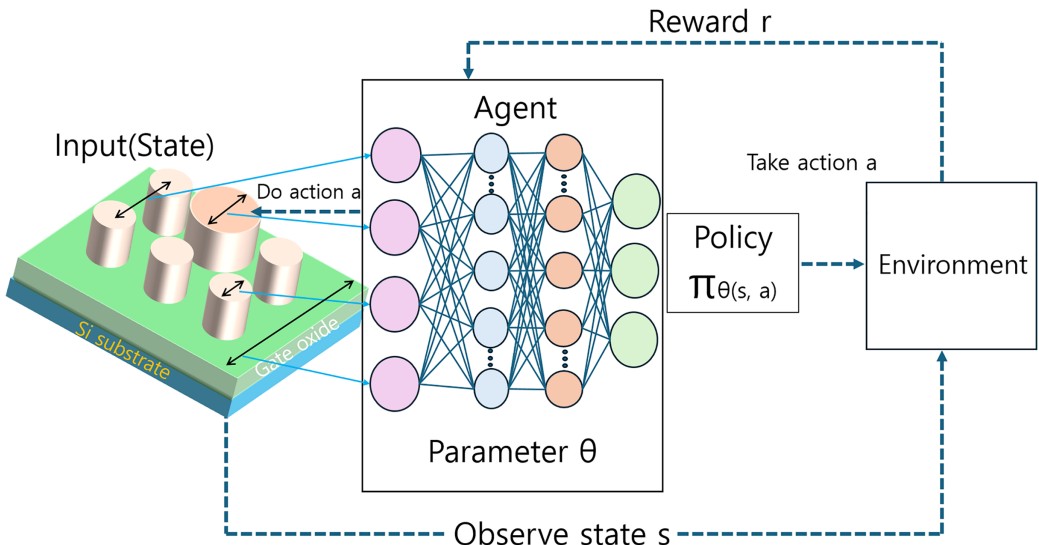

**Figure 7 Schematic representation of a RL framework for optical device inverse design.**

optical functionality—analogous to how a rough sketch might be refined into a detailed blueprint.

- **Robust performance evaluation:** Testing generated designs across parameter variations to ensure performance remains stable despite small fabrication deviations, similar to how engineers build tolerance into mechanical parts.

These practical considerations are essential for bridging the gap between computationally optimized designs and successfully fabricated devices. Recent work by *Zheng et al. (2024)* in flexible electronics fabrication demonstrates how manufacturing constraints can be successfully integrated into the design process, producing devices that maintain performance even under physical deformation.

## Reinforcement learning

RL has emerged as a powerful approach in the inverse design of optical devices, offering unique capabilities in handling complex, sequential decision-making processes. As described by *Sutton & Barto (2018)*, RL models are composed of agents, environments, states, actions, and rewards, creating a framework where an agent learns to execute specific actions through interactions with its environment.

### *Structure and functionality of RL*

In the context of optical device inverse design, RL primarily applies to scenarios where design decisions must be made sequentially, with each decision potentially affecting subsequent choices and outcomes. The objective of RL models in this domain is to guide the agent's behavior towards maximizing cumulative rewards over time, thereby incrementally improving design policies.

Figure 7 illustrates the configuration of an RL system used in the inverse design of optical devices. In this setup, the state represents the current configuration of the optical device and its environment, influencing the agent's action selection. The actions correspond to design choices or parameter adjustments, while the rewards reflect the performance outcomes of these decisions, typically measured in terms of desired optical properties or device efficiency.

### Applications in nanophotonic design

The application of RL in nanophotonic design has shown significant promise, particularly in optimizing complex, multi-parameter systems. Several recent studies have demonstrated the effectiveness of RL-based approaches in this field (*Jiang & Yoshie, 2022*; *Zhao et al., 2022*, *2023*; *Sajedian, Badloe & Rho, 2019*; *Hwang, Lee & Seok, 2022*).

*Autonomous design of nano-thin film devices*
*Jiang & Yoshie (2022)* applied RL to the design of nano-thin film devices, achieving a fully autonomous design process without human intervention. Their approach optimized parameters such as layer thickness and structure size, managing the entire design process from initial concept to final optimization. This work demonstrated the potential of RL in handling continuous feedback-based optimization of challenging parameters, including material selection in thin film design. The ability to autonomously navigate complex design spaces represents a significant advancement in optical device inverse design, potentially reducing the need for extensive human expertise and accelerating the design process.

*Data-efficient optimization strategies*
Building on the concept of combining evolutionary algorithms with RL, *Zhao et al. (2022)* introduced an iterative optimization strategy using deep greedy algorithms. This approach enhanced performance through iterative processes while reducing dependency on large training datasets. The method efficiently performed inverse design for nanophotonics composed of unique material heterojunctions. However, the authors noted a potential risk of overfitting due to limited training datasets, highlighting the importance of balancing data efficiency with model generalization in RL-based design approaches.

### Q-learning for nanophotonic design

*Zhao et al. (2023)* employed a Q-learning algorithm, known for its ability to learn optimal state-value functions in problems involving stochastic transitions and rewards, from which appropriate policies can then be derived. This approach addressed some of the limitations of traditional optimization methods in nanophotonic design. However, the authors acknowledged a known issue with deep Q-learning: the potential for overestimation of Q-values due to the use of the same policy for action selection and evaluation.

To mitigate this issue, *Sajedian, Badloe & Rho (2019)* applied double deep Q-learning (DDQN) in nanophotonics. They conducted simulations on transmission and reflectivity, deriving rewards from the color differences between the resulting and target colors. This

application of DDQN demonstrated the potential for more stable and accurate RL-based optimization in optical device design.

*Scalable RL for multiple design candidates*

Addressing a common limitation in RL-based inverse design, the presentation of only a single design candidate, *Hwang, Lee & Seok (2022)* proposed a deep reinforcement neural network model called Inverse DEsign Agent (IDEA). This model was designed for the creation of 2D optical devices, with the agent trained to produce outcomes for previously unconsidered optical properties. By integrating a tree-based algorithm, IDEA overcame the single-candidate limitation of previous RL methods, generating multiple design candidates and significantly reducing training time.

### Limitations and future directions

Significant promise has been demonstrated in optical device inverse design through RL approaches, though several challenges remain. One key issue is the balance between exploration and exploitation in complex design spaces. While *Jiang & Yoshie (2022)* demonstrated RL's potential in autonomously designing nano-thin film devices, efficiently exploring vast parameter spaces remains difficult. This challenge necessitates more sophisticated exploration strategies for effectively navigating high-dimensional design spaces typical in optical device optimization. Future studies could investigate adaptive exploration techniques that dynamically adjust based on the current state of the design process.

The data efficiency of RL models in optical device design presents another significant challenge. *Zhao et al. (2022)* addressed this by introducing an iterative optimization strategy using deep greedy algorithms, which reduced dependency on large training datasets. However, the risk of overfitting due to limited data persists. Developing data-efficient RL algorithms specifically for optical design applications is crucial for broader adoption. Future work could explore meta-learning approaches enabling RL agents to quickly adapt to new design tasks with minimal data or investigate techniques for generating synthetic training data that accurately represents optical system physics.

Stability and convergence of RL algorithms in optical device design contexts require further investigation. The work of *Zhao et al. (2023)* on Q-learning and *Sajedian, Badloe & Rho (2019)* on DDQN highlighted challenges in ensuring stable learning and avoiding overestimation of action values. Enhancing RL algorithm stability for optical design tasks is essential for producing reliable and consistent results. Future research could explore more robust RL architectures or investigate hybrid approaches combining RL with traditional optimization techniques to ensure more stable convergence to optimal designs.

Current trends indicate a shift towards more sophisticated RL architectures for specific aspects of optical device design. The development of IDEA by *Hwang, Lee & Seok (2022)*, which addresses the limitation of single candidate generation in previous RL methods, exemplifies this trend. This direction suggests a move towards more versatile and scalable RL models in optical device design. Future research is likely to continue this trajectory,

developing increasingly specialized RL architectures that can handle a broader range of optical design problems and generate multiple viable design candidates.

The integration of RL with other computational techniques is an emerging trend. The combination of RL with tree-based algorithms, as seen in *Hwang, Lee & Seok*'s *(2022)* work, suggests a move towards hybrid approaches leveraging the strengths of multiple techniques. Future work could explore more sophisticated combinations of RL with other machine learning models or physics-based simulations to create more comprehensive inverse design frameworks.

Several promising directions for future research in RL-based inverse design of optical devices emerge. Developing physics-informed RL models represents a particularly important avenue. Incorporating known physical laws and constraints directly into RL framework could lead to more realistic and manufacturable design outputs. This approach could involve developing custom reward functions that reflect physical principles or designing state representations that explicitly encode relevant physical parameters.

Extending RL approaches to handle multi-objective optimization tasks is another crucial direction. Many practical optical design problems involve trade-offs between multiple performance criteria. Developing RL frameworks capable of navigating complex multi-objective optimization landscapes could enable the design of optical devices with unprecedented combinations of properties. This could involve the development of novel multi-objective RL algorithms or the adaptation of existing techniques from other fields to specific challenges of optical device design.

The application of RL to dynamic or reconfigurable optical device design represents an unexplored frontier. Most current studies focus on static device structures, but many advanced applications require devices that can adapt to changing conditions. Developing RL models that can generate designs for tunable or actively controlled optical devices could open new possibilities in fields such as adaptive optics or programmable photonics. This would require the development of RL frameworks that can optimize not just initial device structure, but also control policies for dynamic operation.

Addressing the scalability of RL-based approaches to larger and more complex optical systems is crucial for their broader adoption in industrial applications. While current studies have focused on relatively simple optical devices, many practical applications require the design of integrated optical systems comprising multiple components. Developing hierarchical RL architectures capable of designing complex, multi-component optical systems could significantly expand the impact of these techniques in real-world applications. This could involve the development of modular RL approaches that can decompose complex design tasks into manageable sub-problems. Additionally, the interpretability of RL-based design decisions remains a significant challenge in optical device inverse design. While RL models can achieve impressive results, understanding the reasoning behind specific design choices is often difficult. This lack of interpretability can hinder the adoption of RL techniques in scientific and engineering communities where understanding underlying principles is crucial. Future research should focus on developing explainable RL models that provide insights into the decision-making process.

## Practical implementation and recent applications

The transition from computational models to practical fabrication represents a critical step in validating inverse design methodologies. Recent advances in flexible electronics offer valuable insights into practical implementation challenges. For instance, *Zheng et al. (2024)* demonstrated successful fabrication of flexible electrolyte-gated transistors (EGTs) using indium zinc tin oxide (IZTO) nanowires, achieving self-adaptive behavior through a facile electrospinning technique. Their approach maintained excellent electrical properties even under significant bending conditions and achieved high recognition accuracy in neuromorphic applications. This implementation highlights how careful material selection and fabrication processes can address key challenges in translating computational designs to physical devices.

The manufacturing challenges observed in *Zheng et al. (2024)* provide valuable lessons for optical device fabrication. Specifically, they overcame structural integrity issues during the fabrication process by employing a multi-stage heat treatment approach, initial baking at 150 °C followed by UV irradiation and precise annealing at 500 °C with controlled heating rates. Similar methodologies could be applied to address the fragility concerns often encountered when fabricating inverse-designed nanophotonic structures. Additionally, their electron beam lithography approach achieved dimensional accuracy within nanometer ranges, a precision level crucial for many optical applications where performance depends critically on exact structural parameters.

When applied to deep learning inverse design workflows, these practical insights suggest that post-processing steps must be incorporated into the design optimization loop. GANs and AEs particularly would benefit from embedding these fabrication constraints directly into their loss functions or latent space representations. This integration would ensure that generated designs maintain not only optimal optical performance but also structural feasibility under real-world manufacturing conditions.

## EXPERIMENTAL VALIDATION METHODS AND STRATEGIES

### Physical fabrication and experimental validation

*Gostimirovic et al. (2023)* demonstrated comprehensive physical validation of their deep learning approach through actual device fabrication. They employed electron beam lithography to fabricate silicon nanophotonic devices on silicon-on-insulator (SOI) wafers. The fabricated devices were characterized using scanning electron microscopy imaging and optical measurements to analyze structural deviations and performance metrics. Their experimental results validated the effectiveness of their CNN model in correcting fabrication-induced structural variations, demonstrating improved dimensional accuracy and optical performance compared to conventional designs.

*Song et al. (2021)* conducted experimental validation through the fabrication and characterization of silicon-based power splitters. Their fabrication process utilized electron beam lithography and dry etching techniques on SOI wafers. The experimental validation involved measuring transmission spectra of the fabricated devices and comparing them with both simulation predictions and deep learning model outputs. Their results demonstrated successful validation of wavelength and polarization dependencies, with

experimental measurements closely matching the model predictions within acceptable tolerances.

## Simulation-based validation

*Chen et al. (2022)* implemented a rigorous simulation-based validation approach using Lumerical FDTD Solutions. Their physics-augmented deep learning framework was validated against a comprehensive dataset of over 10,000 diverse structural configurations. The validation process focused particularly on electromagnetic field distribution predictions, demonstrating high accuracy in predicting optical responses across various geometric parameters. The simulation results showed excellent agreement between the model predictions and finite-difference time-domain (FDTD) calculations, with average prediction errors below 5%.

*Kojima et al. (2021)* employed a dual-simulation validation strategy, combining FDTD and fine element method (FEM) methods through Lumerical FDTD Solutions and COMSOL Multiphysics. Their validation dataset comprised over 20,000 design variants, extensively testing the model's predictive capabilities across multiple wavelength regimes. The cross-validation between different simulation methods provided robust verification of their deep learning model's accuracy and reliability in predicting optical device performance.

*Shi et al. (2022)* utilized CST Studio Suite for comprehensive electromagnetic simulations, validating their capsule network-based design approach. Their validation methodology encompassed both time-domain and frequency-domain simulations across more than 5,000 nanostructure configurations. The validation process particularly focused on structural symmetry and scaling effects, demonstrating the model's reliability in predicting optical responses. Their simulation results showed strong correlation between predicted and simulated performance metrics, with mean squared errors consistently below predetermined thresholds.

## Comparative analysis case of deep learning techniques

The investigation of deep learning techniques for color generation in nanophotonic devices reveals nuanced capabilities across different methodological approaches. Drawing from the pioneering work of *Gao et al. (2019)*, MLP architectures demonstrate remarkable potential in silicon-based color design, capable of generating over a million distinct colors. However, their performance critically depends on the comprehensiveness and diversity of training datasets.

Generative Adversarial Networks (GANs) present a complementary approach, offering unprecedented capabilities in generating novel nanophotonic structures with unique color-generating properties. These models excel in navigating complex design spaces, potentially discovering color production mechanisms not previously conceived through traditional design methodologies.

The inherent trade-offs between these techniques become evident when examining their specific characteristics. MLPs offer moderate computational efficiency but face limitations in handling highly complex design scenarios. Conversely, GANs can manage intricate

**Table 3 Comparison of deep learning methods in nanophotonic color design.**

| Method | Color generation | Computational efficiency | Design complexity |
|--------|------------------|--------------------------|-------------------|
| MLP | Million colors | Moderate | Limited |
| GAN | Novel designs | Low | High |

structural designs but encounter significant challenges in ensuring the physical realizability of generated nanostructures.

This comparative analysis underscores a critical insight: the selection of deep learning techniques in nanophotonic color generation must be meticulously aligned with specific application requirements, computational constraints, and desired design outcomes. The evolving landscape of inverse design methodologies demands a nuanced understanding of each technique's unique strengths and limitations.

Experimental validation remains a crucial frontier, bridging the gap between computational design and practical implementation. Future research should prioritize comprehensive validation strategies that can effectively translate these advanced computational approaches into manufacturable optical devices with precise color-generating capabilities.

Table 3 represents of sample such as different deep learning methods demonstrate varying capabilities in nanophotonic color design.

## DISCUSSION

This review analyzed recent deep learning methodologies for the inverse design of nanophotonic devices. Undertaking a study to systematically design and validate each deep learning model through to the actual production of devices presents significant challenges. The investigation into inverse design for nanophotonic devices utilizing deep generative models is significant, as it allows for the evaluation of the suitability of various deep learning models in the design of specific devices.

Our analysis indicates that MLP networks have been widely used in early research due to their straightforward implementation and ability to model complex nonlinear relationships between device parameters and optical properties. Studies such as those by Liu et al. (2022, 2023) demonstrated the use of MLPs in designing quantum nanoparticle shells, addressing challenges like shell thickness and electromagnetic environment interactions. However, MLPs often require large datasets and may struggle with scalability as device complexity increases.

CNNs have shown potential in handling spatial data inherent in optical device design, particularly for 2D and 3D structures. Researchers like Song et al. (2020, 2021) utilized CNNs to manage various parameters, including wavelength and polarization, enhancing computational efficiency in the inverse design process. However, the application of CNNs has been limited, possibly due to the scarcity of image data representing optical devices and the computational cost associated with training deep CNN architectures.

AEs, including their variants like VAEs and CVAEs, have been effective in addressing the curse of dimensionality in nanophotonic design. By reducing the dimensionality of complex design spaces, AEs facilitate efficient inverse design processes. Studies by *Li et al. (2022)* and *Tang et al. (2020)* highlighted the utility of AEs in unsupervised learning-based inverse design, although challenges remain in integrating these models with conventional electromagnetic analysis tools and verifying the manufacturability of the generated designs.

GANs have been applied to generate novel nanophotonic structures with desired optical properties. *Kim et al. (2022)* introduced a GAN-based inverse design method, incorporating conditional GANs with controllable classifiers to achieve specific design outcomes. While GANs show promise in generating diverse designs, they can be difficult to train due to instability in the adversarial process and may require large amounts of data.

RL approaches have been explored for autonomous and sequential decision-making in optical device design. *Jiang & Yoshie (2022)* demonstrated the use of RL for fully autonomous design of nano-thin film devices, optimizing parameters like layer thickness and material selection. RL methods can effectively handle complex optimization tasks but may suffer from overfitting due to limited training data and require careful design of reward functions to guide the learning process appropriately.

Although all five methods are currently in use, MLPs have been the most frequently employed, with AEs and RL also seeing considerable application. The application of GANs appears to be less prominent, possibly due to the complexities in training and the advancements of other generative models. Similarly, the use of CNNs has been limited, which may be attributed to the challenges in acquiring sufficient image data and the computational resources required.

A common limitation across these methods is the need for large, high-quality datasets to train the models effectively. Data scarcity can lead to overfitting and limit the generalizability of the models to new design spaces. Additionally, integrating these deep learning models with existing simulation tools and experimental validation processes remains a challenge. Future research could focus on leveraging state-of-the-art architectures such as Transformers (*Vaswani et al., 2017*) and diffusion models (*Ho, Jain & Abbeel, 2020*), which have shown remarkable capabilities in other domains.

While research on applying Transformers and diffusion models to inverse design of optical devices is currently limited, their remarkable success in computer vision and natural language processing suggests significant potential for nanophotonic applications. Optical device design involves complex interactions between various parameters including structural geometry, material properties, and wavelength characteristics. The self-attention mechanisms in Transformers enable learning relationships between all these elements simultaneously, allowing the model to effectively capture complex nonlinear relationships that are crucial for device optimization. Additionally, the iterative refinement process inherent in diffusion models presents a promising approach for generating high fidelity optical device designs, as their denoising methodology aligns well with the need for precise structural control in nanophotonics. The proven ability of these models to handle high-dimensional data and generate diverse yet physically

plausible outputs makes them particularly appealing for next generation inverse design systems.

By incorporating these advanced models, it may be possible to develop novel inverse design methodologies that surpass current techniques. Furthermore, adopting a 3D structural perspective in the inverse design process, particularly utilizing advancements in generating high-fidelity 3D images, holds promise for enhancing device design efficiency. This approach could streamline the design process in the pre-production phase compared to traditional 2D methodologies. Finally, integrating the inverse design process with actual fabrication and experimental validation is crucial. Follow-up studies that include the validation of inverse-designed nanophotonic devices through fabrication-level processes are needed to ensure that the designs produced by these models are practically realizable and can meet the desired performance criteria in real-world applications.

## CONCLUSION

This review has explored the recent advancements in the inverse design of optical devices using deep generative models. The integration of deep learning techniques into the design process has opened new avenues for creating optical devices with desired properties, significantly enhancing efficiency and expanding the range of possible designs.

While this review focused on device-level inverse design, integrating inverse design across multiple scales could revolutionize optical device development. Recent advances in molecular design (_Anstine & Isayev, 2023_; _Mroz et al., 2022_), and materials discovery (_Sanchez-Lengeling & Aspuru-Guzik, 2018_) using machine learning suggest promising opportunities for a unified multi-scale inverse design framework. Such an approach could enable simultaneous optimization of material properties and device architectures, potentially leading to unprecedented device performance and functionality. This holistic strategy represents an exciting direction for next-generation optical devices.

The field of inverse design in nanophotonics stands at the cusp of a transformative era. The rapid evolution of deep generative models, particularly Transformers (_Vaswani et al., 2017_) and diffusion models (_Ho, Jain & Abbeel, 2020_), presents exciting opportunities for addressing current limitations in optical device design. While these models have demonstrated remarkable capabilities in natural language processing and image generation, their potential in nanophotonics remains largely untapped. Adapting the principles of Transformers to optical device design could revolutionize how we capture and process complex relationships in nanophotonic structures. The self-attention mechanisms inherent in Transformer architectures might enable more sophisticated modeling of long-range interactions within optical devices, potentially leading to designs that exploit subtle physical phenomena currently overlooked by traditional methods. Diffusion models, with their unique approach to generative modeling, could offer novel strategies for producing high-fidelity device designs. Their iterative refinement process might be particularly well-suited to navigating the complex design spaces encountered in nanophotonics, potentially yielding more stable and reliable design outcomes. The application of these models to inverse design tasks could significantly

enhance our ability to generate diverse, yet physically realizable, optical device configurations.

A promising direction for future research lies in the adoption of three-dimensional structural perspectives in the inverse design process. By leveraging advancements in generating high-fidelity 3D representations, as demonstrated in fields like computer vision (*Mildenhall et al., 2021*; *Kerbl et al., 2023*), we could dramatically enhance the efficiency and accuracy of device design. This approach could enable more comprehensive modeling of optical phenomena in complex structures, potentially streamlining the pre-production phase compared to traditional two-dimensional methodologies.

The integration of deep learning-based inverse design with fabrication and experimental validation processes represents a critical frontier. Ensuring that generated designs are not only theoretically optimal but also practically realizable and performant in real-world applications is paramount. This challenge necessitates close collaboration between computational scientists and experimental physicists, bridging the gap between theoretical designs and practical implementations.

## ABBREVIATION

Table 4 introduces an Abbreviations Table as attached to clarify technical terms.

**Table 4 Abbreviations and expansions in optical device and deep learning terminology.**

| Abbreviation | Expansion | Meaning |
| --- | --- | --- |
| MLP | multi-layer perceptron | Neural network with multiple hidden layers |
| CNN | convolutional neural network | Deep learning model for processing spatial data |
| AE | auto-encoder | Neural network for data compression and reconstruction |
| VAE | variational auto-encoder | AE variant with probabilistic encoding |
| CVAE | conditional variational auto-encoder | VAE with additional conditional information |
| GAN | Generative Adversarial Network | Generative model with competing neural networks |
| RL | reinforcement learning | Machine learning approach based on reward-driven learning |
| DDQN | double deep Q-learning | Advanced Q-learning technique |
| IDEA | Inverse DEsign Agent | Deep reinforcement learning model for design |
| FDTD | finite-difference time-domain | Computational electromagnetic simulation method |
| FEM | finite element method | Numerical technique for solving complex engineering problems |
| SOI | silicon-on-insulator | Semiconductor wafer technology |
| EM | electromagnetic | Relating to electric and magnetic field interactions |
| DBR | distributed Bragg reflector | Optical mirror constructed from multiple dielectric layers |
| LED | light emitting diode | Semiconductor device emitting light |
| OLED | organic light emitting diode | Light-emitting diode using organic semiconductors |
| MEMS | micro-electro-mechanical systems | Microscale mechanical and electronic devices |
| NIR | near-infrared | Electromagnetic radiation wavelength range |
| QD | quantum dot | Nanoscale semiconductor particle |
| PCW | photonic crystal waveguide | Optical waveguide using periodic optical structure |

## ACKNOWLEDGEMENTS

The authors acknowledge the use of Writefull, an AI-powered language editing tool, for grammar and spelling checks in the manuscript preparation process.

### Funding

This work was supported by the National Research Foundation of Korea (NRF) grant funded by the Korea government (MSIT) (RS-2024-00337250), and by Korea Institute for Advancement of Technology (KIAT) grant funded by the Korea Government (MOTIE) (P0020967, Advanced Training Program for Smart Sensor Engineers.) The funders had no role in study design, data collection and analysis, decision to publish, or preparation of the manuscript.

### Grant Disclosures

The following grant information was disclosed by the authors:
National Research Foundation of Korea (NRF).
Korea Government (MSIT): RS-2024-00337250.
Korea Institute for Advancement of Technology (KIAT).
Korea Government (MOTIE): P0020967.

### Competing Interests

The authors declare that they have no competing interests.

### Author Contributions

- Sun Jae Baek conceived and designed the experiments, performed the experiments, analyzed the data, prepared figures and/or tables, authored or reviewed drafts of the article, and approved the final draft.
- Minhyeok Lee conceived and designed the experiments, authored or reviewed drafts of the article, and approved the final draft.

### Data Availability

   This is a literature review.

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
