# Peer review of "Recent advances in the inverse design of silicon photonic devices and related platforms using deep generative models"

_PeerJ Computer Science, doi:10.7717/peerj-cs.2895_

## Round 0.1 · original submission · Major Revisions

Dear Authors,

The manuscript provides a thorough review of recent advancements in applying deep generative models to optical device design and aligns well with the journal's scope. However, significant revisions are needed to improve clarity, logical flow, and the practical relevance of the review, particularly in addressing experimental validation and enhancing figures. These improvements are essential to maximize the manuscript's impact and ensure it meets PeerJ's high publication standards.

Please, answer all the points raised by ALL the Reviewers

Best,

M.P.

Reviewer 1 ·

Basic reporting

The authors state that "traditional design approaches often struggle with these challenges, highlighting the need for more sophisticated techniques," and then transition to discussing the need for deep generative models. While this is a valuable perspective, it is important to note that in the field of optics and photonics, methods such as adjoint optimization, topology optimization, and inverse design have already addressed many of these challenges with significant success. In fact, the term "inverse design" is often closely associated with adjoint and topology optimization within the optics community.

I would kindly suggest that the authors consider including a brief discussion of these approaches, as they are highly relevant to the context of the paper. For instance, the following articles provide useful insights: Optics Express 28.5 (2020): 6945-6965 and ACS Photonics 7.8 (2020): 2236-2243. Additionally, there has been promising work on combining adjoint optimization with neural networks, such as the study in Materials & Design 239 (2024): 112737. Including these references would not only broaden the scope of the discussion but also help position the work within the wider context of existing optimization techniques in the field.

Experimental design

No comment

Validity of the findings

No comment

Reviewer 2 ·

Basic reporting

This review article examines recent advancements in using deep generative models for the inverse design of optical devices. The authors provide a comprehensive overview of various deep learning methods, including multilayer perceptrons (MLP), convolutional neural networks (CNN), autoencoders (AE), generative adversarial networks (GAN), and reinforcement learning (RL). The article analyzes the application of these techniques to the inverse design of optoelectronic devices, comparing their effectiveness and integration into design processes. It also explores the limitations of these approaches and suggests future research directions.
While the manuscript offers a thorough overview of deep generative models, encompassing MLP, CNN, AE, GAN, and RL architectures, there are several areas requiring improvement before publication:
Specific Recommendations:
1. Include Necessary References: Ensure all relevant references are cited in relation to Table 1.
2. Expand High-Impact Journal Comparisons: Incorporate more high-impact journals, such as Nature Photonics and Laser & Photonics Reviews, in Table 2 for broader and more meaningful comparisons.
3. Clarify MLP Scale and Structure: Provide details about the structure size or scale range of Multi-Layer Perceptrons (MLPs) as depicted in Fig. 2, and mention this explicitly in the text.
4. Deepen Analysis of Advanced Models: Certain sections are overly descriptive and lack detailed analysis. For example, the discussions on Transformers and diffusion models are relatively brief, despite these models' significant success in other fields.
5. Broaden Scope: The article predominantly focuses on silicon-based nanophotonics, with limited discussion of other material platforms or optical device types. Expanding the scope to include metamaterials, optical fibers, and quantum optical devices would enhance the article’s impact and relevance.
6. Address Experimental Validation: The article briefly mentions the importance of integrating deep learning models with fabrication processes but lacks specific guidance on validating the performance of generated designs. Include a section discussing experimental validation, including challenges and strategies for translating designs into manufacturable devices.
7. Improve Organization: The article’s organization could be refined, as transitions between sections sometimes feel abrupt. Ensure a smoother flow to enhance readability and logical progression.

Experimental design

no comment

Validity of the findings

no comment

Additional comments

Actionable Suggestions (revise them as posible):
1. Expand Discussion of Transformers and Diffusion Models: Provide more detailed insights into how these models can be applied to the inverse design of optical devices.
2. Explore Applications Beyond Silicon: Discuss the potential of deep learning methods for designing devices in other material platforms and categories, such as metamaterials, optical fibers, and quantum optics.
3. Highlight Experimental Integration: Add a dedicated section on experimental validation, addressing how designs can be tested and implemented in real-world fabrication processes.
4. Enhance Logical Flow: Reorganize sections to ensure smoother transitions and a clearer structure.

By addressing these points, the manuscript can provide a more comprehensive and impactful overview of using deep generative models for the inverse design of optical devices.

Reviewer 3 ·

Basic reporting

The manuscript is written in professional English, but some sections are overly dense and technical, potentially limiting accessibility for readers less familiar with deep learning or optical device design. Simplifying key areas and clarifying technical terms would improve clarity. The literature review is comprehensive, with sufficient background and context provided, referencing recent advancements and positioning the study within the field. However, it could better highlight existing gaps in the literature to strengthen its relevance. The article adheres to a logical structure with well-organized sections, but figures and tables could be more detailed to enhance understanding; for instance, annotated diagrams and a comparative summary table of methodologies would be beneficial. While raw data is not applicable to this review, the systematic categorization of studies ensures verifiability and usability. The topic is of broad, cross-disciplinary interest, aligning with the journal’s scope, but the manuscript could better frame its contributions for non-specialist readers. It justifies its novelty by focusing on the most recent advancements (2019–2024) and offering a comprehensive perspective on deep generative models applied to optical design, filling a gap in the existing reviews. The introduction provides sufficient background and motivation but could further clarify the intended audience and practical implications of the research. While formal definitions and theorems are not required, the technical descriptions of certain methods, such as GANs and reinforcement learning, would benefit from greater detail and illustrative examples to ensure broader comprehension.

Experimental design

The manuscript fits well within the aims and scope of the journal as a literature review, providing a detailed and technical analysis of recent advancements in applying deep generative models to the inverse design of optical devices. The review demonstrates a rigorous approach, with a high standard of technical accuracy and ethical conduct in analyzing and synthesizing the literature. However, while the methodologies of the reviewed studies are described in detail, certain sections could provide additional explanations or examples to ensure full reproducibility and enhance the manuscript's practical utility.
The survey methodology appears consistent with a comprehensive and unbiased approach, as it covers a wide range of deep learning techniques and their applications in optical device design. However, some areas could benefit from greater elaboration, such as the rationale behind the selection of specific studies and a more explicit discussion of potential biases or limitations in the reviewed literature. Sources are adequately cited, with quotations or paraphrasing used appropriately, but highlighting the most influential studies in a dedicated section would help emphasize their impact on the field.
The review is generally well-organized into coherent paragraphs and subsections, with clear categorizations of methodologies and applications. That said, the addition of a high-level summary at the end of each section to synthesize findings and connect them to overarching trends would further enhance readability and logical flow. Including more detailed subsections for emerging techniques, such as conditional generative models or hybrid approaches, could also improve the review's structure and depth.

Validity of the findings

The manuscript does not assess the impact and novelty of the findings directly, as it is a review article, but it provides a comprehensive synthesis of recent advancements, which lays a strong foundation for future research in the field. The review encourages meaningful replication by presenting a detailed analysis of methodologies and their applications, along with a clear rationale and benefits for advancing the literature. However, providing more explicit examples of studies that can be replicated or extended would strengthen its utility further.
The conclusions are well stated and effectively linked to the original research question, summarizing the key findings and staying focused on the supporting results. The arguments developed in the manuscript are well-supported and align with the goals outlined in the introduction, which aims to provide a comprehensive overview of deep generative models for the inverse design of optical devices.
The conclusion identifies several unresolved questions and gaps, such as the need for improved experimental validation, the scalability of models, and the integration of physical constraints into deep learning frameworks. It also highlights future directions, including the development of more interpretable models, multi-objective optimization approaches, and applications to dynamic optical devices. These insights provide a clear pathway for advancing research in this interdisciplinary field. However, expanding on the practical implications of addressing these gaps and providing specific suggestions for experimental work would further enhance the manuscript's impact.

Additional comments

Manuscript #106582 Review comments

The manuscript provides a comprehensive literature review on the use of deep generative models for the inverse design of optical devices. It explores various methodologies, including Multi-Layer Perceptrons (MLPs), Convolutional Neural Networks (CNNs), Autoencoders (AEs), Generative Adversarial Networks (GANs), and Reinforcement Learning (RL). The authors analyze these methods' applications, strengths, and limitations in the design of photonic devices, particularly focusing on their ability to navigate complex parameter spaces and optimize device performance. The paper concludes with a discussion on the trends, challenges, and future directions in applying deep learning to optical device design. The following are the major revision suggestions for this manuscript:

1. The manuscript would benefit from a more structured connection between data analysis and the implications for the field of optical device design. For example, the discussion on GANs highlights their potential but does not adequately link this to their practical utility in manufacturing optical devices. Providing clearer transitions and explicitly stating how each challenge (e.g., scalability, interpretability) ties to proposed solutions or future research directions would improve the manuscript's logical flow and make the content more engaging and insightful for readers.
2. While the manuscript dives deeply into the capabilities of specific deep learning techniques like MLPs, CNNs, GANs, and RL, it lacks a holistic comparison of their suitability for different optical design scenarios. For instance, instead of treating each method as an isolated case, the authors could include a comparative analysis summarizing when and why a particular technique might outperform others in certain applications. This could be effectively presented in a summary table to provide clarity and assist readers in identifying the best method for their specific needs.
3. The paper provides an extensive review of theoretical advancements but does not discuss the experimental realization of designs generated by these deep learning models. This omission weakens the manuscript’s relevance to practical applications. Including examples of successful implementations or discussing the barriers preventing physical realization, such as fabrication limitations or discrepancies between simulations and real-world performance, would enhance the paper's impact and provide a more balanced perspective.
4. The manuscript's dense and highly technical language could alienate readers unfamiliar with deep learning or optical device design. For instance, complex descriptions of neural network architectures lack accompanying simplifications or analogies that would make the content more accessible. Simplifying the language, particularly in sections discussing technical methodologies, and providing additional context or glossaries for less-experienced readers would significantly broaden the paper's audience.
5. Some figures, such as model schematics, are too simplistic or lack sufficient detail to fully convey the described processes. For instance, diagrams illustrating GAN training or reinforcement learning systems do not include enough information about critical parameters or workflows. Enhancing these visuals with annotated details or providing additional explanatory figures for more complex methodologies would improve readers' understanding and retention of key concepts. Clearer and more comprehensive visuals would align better with the manuscript's informative intent.

The manuscript presents a valuable and timely review of an important topic; however, it requires major revisions to reach the necessary standards for publication. Key improvements are needed in logical coherence, the integration of practical relevance, and the overall clarity of presentation. By addressing these critical issues, the manuscript can significantly enhance its impact, better serve its target audience, and effectively contribute to advancements in the field.

Annotated reviews are not available for download in order to protect the identity of reviewers who chose to remain anonymous.

Reviewer 4 ·

Basic reporting

see additional comment

Experimental design

see additional comment

Validity of the findings

see additional comment

Additional comments

The submitted review by Baek and Lee provides a high-level overview of artificial intelligence techniques for inverse design of optical devices. I believe the Authors’ have amassed a significant collection of relevant references to discuss. The topic is certainly of interest to the readership. However, in its current form I believe publication would be premature. While the article is stylistically well-constructed, significant details and discussion surrounding the references discussed need to be provided. I do not go through and list every instance in detail, but this is the case for the majority of references in the manuscript. As it stands, the Authors’ have compiled and presented a list of published works, but they do not outline these findings (or methods improvement) and place them in the greater context of the review. Moreover, the figure design is repetitive and mostly uninformative for the purposes of inverse design. These should be remade to publication quality and content prior to acceptance. Overall, I assert that the article will be suitable for acceptance only after major revision.
General Comments
• The portion of the Review that focuses specifically on MLPs should be expanded to provide additional depth to the reader. As an example, for some of the highlighted studies the Author’s should indicate what the input parameters are to the MLP predictors and then discuss what they Author’s were seeking to optimize / were they successful. Descriptions or insight into the perceived causes of these successes / failures should be included. As it stands, this Section of the review appears mostly as a listing of studies.
• “As observed in the work of Liu et al. (2022), the error rate increased with growing shell thickness in quantum nanoparticle designs. This indicates that as device structures become more intricate, the computational demands for MLP-based inverse design grow substantially. Future research could focus on developing more efficient network architectures or training methods to address this scalability issue, potentially incorporating techniques from other deep learning domains such as sparse or quantized neural networks.” I do not believe the Authors’ proposed future work resolves the deficiency that is highlighted. This appears to be a data scarcity problem as opposed to a neural network scalability problem. Reduced precision / quantized training can provide some benefit to scaling the MLP training, but I would be very surprised if this is actually a compute saturated problem.
• Figure 3 is not informative for the reader. It is not clear what the convolution is meant to be operating on in this image. The figure should be redesigned.
• There are some locations in the text where informal or imprecise language is used. For instance, “While CNNs demonstrate high predictive accuracy, understanding their decision-making process can be challenging.” CNN’s themselves do not “make decisions.” They provide predictions and the scientists are meant to utilize that predictions to make design decisions. The exception to this would be in a reinforcement learning set up, where the CNN serves as an RL agent. In the next sentence “This lack of transparency may hinder the adoption of CNN-based methods in scientific applications where understanding underlying physics is crucial.” This lack of transparency is not specific to CNNs, but all high-dimensional data-driven methods discussed.
• Figure 5 should be remade. The figure is overly simplistic and ultimately it does not show how GANs are used in inverse design (the topic of the review).
• In the Q-learning section “Zhao et al. (2023) employed a Q-learning algorithm, known for its ability to converge to optimal action values in problems involving stochastic transitions and rewards.” Q-learning does not learning action-values because it is not a policy iteration method. It either learns the state-value function or the state-action-value function, from which a policy needs to be applied on top of.
Stylistic comments:
• The choice to include a specific section detailing how and where articles were obtained for the Review is, to me, an atypical choice. I am unsure if this is a typical fixture in a PeerJ Comp. Sci. Review article, but if it is not, I recommend removing it. As a minimum alternative, it could be shortened to just include Table 1 and Table 2. Figure 1 is mostly uninteresting for the readership. Paragraph 1 and the first half of paragraph 2 can be deleted. The second half of paragraph 2 can be merged into the remaining text.
• “One potential approach could be the integration of physics-informed neural networks, which incorporate known physical laws into the learning process, potentially enhancing the model’s ability to generalize beyond the training data.” The term “potential” is repetitive in this sentence.
• “The RL” should just be RL.
Textual comments:
• “The limitation of tandem neural networks in addressing only single-solution problems has been addressed by integrating particle swarm optimization techniques.” I am not sure what is meant by “tandem neural networks.”
• Similarly, “Guo et al. (2022) introduced the concept of an extendable neural network” I am not sure what is meant by an extendable neural network. These terms should be described for the reader.
• "Unlike conventional MLP models that rely solely on backpropagation during training, this method optimizes the geometric structure of photonic devices in a polar coordinate system, even with relatively limited training datasets." This is an example where more depth is needed for the reader. How is the MLP being used here? Is it being used as a fitness function? Mutation engine? Population generator?
• “The application of CNN architectures in the inverse design of silicon-based nanophotonics has been demonstrated at both the process level for feedback and the design level for simulation scenarios. Several recent studies have showcased the effectiveness of CNNs in various aspects of nanophotonic design (Gostimirovic et al., 2023; Song et al., 2021, 2020; Chen et al., 2022; Shi et al., 2022; Ma et al., 2022).” Is another example where details and discussion are needed. These statements don’t provide much value beyond “there have been studies done by people.” (in simplistic direct terms).
• “Future research could investigate techniques for extracting physical insights from trained GAN models, possibly through the integration of attention mechanisms or the development of hybrid models that combine GANs with more interpretable machine learning approaches.” It is unclear how attention would help with interpretability. Presumably the Authors are referring to graph attention or convolutional attention, but without context the reference here is peculiar.
General recommendation:
It is perhaps relevant to include a few comments in the conclusion about the related molecular and material inverse design efforts. The current Review focuses mainly on device fabrication / configuration, however, one can easily imagine that inverse design at all scales for optical devices can readily form a cohesive strategy for a new generation of devices. Relevant papers to cite in these fields would be:
[1] https://doi.org/10.1021/jacs.2c13467
[2] https://doi.org/10.1021/jacs.2c06833
[3] https://doi.org/10.1126/science.aat2663

---

## Round 0.2 · Minor Revisions

Dear Authors,

please implement the suggestions given by Reviewer 3 regarding the Experimental Design and the Validity of the findings.


Regarding the inclusion of the paper, you are free to add it or not.
While the implementation of the above suggestion is mandatory, you have free of choice for the paper.

Best regards,
M.P.

Reviewer 1 ·

Basic reporting

The manuscript is now well-written and ready to be published.

Experimental design

The manuscript is now well-written and ready to be published.

Validity of the findings

The manuscript is now well-written and ready to be published.

Additional comments

The manuscript is now well-written and ready to be published.

Reviewer 2 ·

Basic reporting

The paper has been thoroughly revised and is now suitable for publication.

Experimental design

NA

Validity of the findings

NA

Additional comments

NA

Reviewer 3 ·

Basic reporting

This study provides a comprehensive review of the inverse design of optical devices using deep generative models. The paper thoroughly discusses various deep learning techniques, including MLP, CNN, AE, GAN, and RL, and their applications in nanophotonic design. The authors offer a detailed analysis of the advantages and limitations of each approach, presenting an insightful synthesis of recent advancements in the field. The study is of significant academic value, particularly in its comparative analysis of different deep learning methods and their practical implications for optical device design.
The authors have addressed the reviewers' previous comments with substantial revisions, including the addition of relevant references, improvements in experimental descriptions, and enhancements to the introduction and conclusion sections. Overall, the paper has reached a high standard, and I recommend it for acceptance after minor revisions.

Experimental design

The discussion on Auto-encoder (AE) in presents its role in optical device design, particularly in "dimensionality reduction." However, the mathematical definition and its specific impact on inverse design could be further clarified. It is recommended that the authors provide a more precise explanation or an example to help readers better understand the role of AE in optical device optimization.

Validity of the findings

The section on GAN applications emphasizes its generative capability but lacks a discussion on potential challenges such as artifacts and training instability. The authors should consider adding a brief discussion on these issues and possible solutions, such as the use of Wasserstein GAN (WGAN) to improve training stability or incorporating physics-based constraints to enhance the physical feasibility of generated designs.

Additional comments

1. Some references do not fully comply with the journal's formatting requirements. For instance, Reference is missing a DOI, and does not follow the correct author formatting. The authors should carefully check all references and ensure they align with the journal’s submission guidelines.
2. The paper presents an insightful comparison of deep learning methods (e.g., CNN vs. MLP) for optical device design. However, the comparative analysis could be further strengthened by including performance metrics such as computational complexity and convergence speed. Additionally, incorporating quantitative results where applicable would further support the paper’s conclusions.Moreover, in the field of adaptive electronic device design, deep learning has been successfully applied to inverse design and optimization. For instance, Zheng et al. (2024) investigated a flexible electrolyte-gated transistor (EGT) based on InZnSnO nanowires, demonstrating its advantages in self-adaptive signal modulation and neuromorphic computing(DOI: 10.1016/j.apmt.2024.102424). The authors may refer to this study to strengthen their discussion on the optimization capabilities of different deep learning models for photonic and optoelectronic systems, as well as their potential applications in future research.

---

## Round 0.3 · Minor Revisions

The authors have adequately addressed most of previous comments, and the revised manuscript now meets the minor revision criteria. There are no additional concerns regarding ethical issues or potential misconduct. please, Address the suggested minor points provided by reviewer 3.

Please, add the citation if and only if it is valuable for you.

Best

Reviewer 3 ·

Basic reporting

This manuscript systematically reviews recent progress in deep generative models (including MLP, CNN, AE, GAN, and RL) for the inverse design of silicon-based nanophotonic devices. The language is clear, and the manuscript's logical structure is well-organized. The background information provided is thorough. However, some sections describing the technical details and model architectures remain somewhat complex. I recommend further simplifying these sections by incorporating intuitive diagrams or analogies to enhance accessibility for non-specialist readers.

Experimental design

The authors have clearly outlined their literature selection criteria and provided a structured review framework, ensuring comprehensiveness and objectivity. The inclusion of a comparative summary table (Table 1) significantly improves readability and enhances logical coherence. However, discussions regarding practical implementations and experimental validations remain relatively superficial. To enhance this, the authors might consider consulting recent literature on practical applications involving flexible electrolyte-gated transistors with self-adaptive capabilities, particularly studies utilizing metal oxide nanowires published recently in Applied Materials Today (e.g., DOI: 10.1016/j.apmt.2024.102424). Integrating insights from such references could significantly strengthen the manuscript’s practical relevance and utility.

Validity of the findings

The manuscript effectively identifies existing research gaps and future research directions, such as scalability and multi-objective optimization. However, the discussions regarding the practical feasibility and physical realizations of deep learning models (especially GANs and AEs) remain somewhat general. It is recommended that the authors clearly specify particular challenges encountered during manufacturing, propose potential solutions, and elaborate more explicitly on issues such as model generalization and interpretability.

Additional comments

The authors have substantially improved the manuscript in response to the previous review. To further enhance the manuscript, please consider the following minor adjustments: (1) Include 1-2 specific examples or successful experimental cases to highlight practical applications clearly; (2) Simplify technical language and add intuitive explanations to improve readability for a broader audience; (3) Expand detailed discussions on potential manufacturing challenges of deep learning models by citing recent practical examples from the literature, such as the work by Zheng et al. mentioned above.

Reviewer 4 ·

Basic reporting

I believe after the Author's changes, the article meets the standards for acceptance for a PeerJ Comp. Sci. article.

Experimental design

I believe after the Author's changes, the article meets the standards for acceptance for a PeerJ Comp. Sci. article.

Validity of the findings

I believe after the Author's changes, the article meets the standards for acceptance for a PeerJ Comp. Sci. article.

Additional comments

The only additional comment I have is a recommendation to the Author's for future submissions. It is convenient for the Reviewer's if every Reviewer comment, no matter how small, is addressed with a point-by-point response. In this case it was somewhat straightforward to identify my comments had been addressed, but the process could've been much more streamlined.

---

## Round 0.4 · accepted · Accept

The authors have addressed all of the reviewers' comments.